# PP1 phosphatase controls both daughter cell formation and amylopectin levels in *Toxoplasma gondii*

**Asma Sarah Khelifa**[1◉], **Maanasa Bhaskaran**[1◉], **Tom Boissavy**[1], **Thomas Mouveaux**[1], **Tatiana Araujo Silva**[2], **Cerina Chhuon**[3], **Marcia Attias**[2], **Ida Chiara Guerrera**[3], **Wanderley De Souza**[2], **David Dauvillee**[4], **Emmanuel Roger**[1], **Mathieu Gissot**[1] *

**1** Univ. Lille, CNRS, Inserm, CHU Lille, Institut Pasteur de Lille, U1019—UMR 9017—CIIL—Center for Infection and Immunity of Lille, Lille, France, **2** Laboratory of Celullar Ultrastructure Hertha Meyer, Biophysics Institute Carlos Chagas Filho, Federal University of Rio de Janeiro, Rio de Janeiro, Brazil, **3** Proteomics platform 3P5-Necker, Université Paris Descartes—Structure Fédérative de Recherche Necker, INSERM US24/CNRS, UMS3633, Paris, France, **4** UGSF—Unité de Glycobiologie Structurale et Fonctionnelle UMR 8576, Lille, France

◉ These authors contributed equally to this work.

* mathieu.gissot@pasteur-lille.fr

**Data Availability Statement:** The mass spectrometry proteomics data have been deposited to the ProteomeXchange Consortium via the PRIDE partner repository with the dataset identifier

## Abstract

Virulence of apicomplexan parasites is based on their ability to divide rapidly to produce significant biomass. The regulation of their cell cycle is therefore key to their pathogenesis. Phosphorylation is a crucial posttranslational modification that regulates many aspects of the eukaryotic cell cycle. The phosphatase PP1 is known to play a major role in the phosphorylation balance in eukaryotes. We explored the role of TgPP1 during the cell cycle of the tachyzoite form of the apicomplexan parasite *Toxoplasma gondii*. Using a conditional mutant strain, we show that TgPP1 regulates many aspects of the cell cycle including the proper assembly of the daughter cells' inner membrane complex (IMC), the segregation of organelles, and nuclear division. Unexpectedly, depletion of TgPP1 also results in the accumulation of amylopectin, a storage polysaccharide that is usually found in the latent bradyzoite form of the parasite. Using transcriptomics and phospho-proteomics, we show that TgPP1 mainly acts through posttranslational mechanisms by dephosphorylating target proteins including IMC proteins. TgPP1 also dephosphorylates a protein bearing a starch-binding domain. Mutagenesis analysis reveals that the targeted phospho-sites are linked to the ability of the parasite to regulate amylopectin steady-state levels. Therefore, we show that TgPP1 has pleiotropic roles during the tachyzoite cell cycle regulation, but also regulates amylopectin accumulation.

## Introduction

Apicomplexa is a phylum that comprises single-celled, obligate, intracellular protozoan parasites. Within this phylum, there are several species of human pathogens, such as *Plasmodium* spp. (the causative agent of malaria), *Toxoplasma* (which causes toxoplasmosis), and

PXD043539. The RNA-seq data was deposited to the SRA database under the identifier PRJNA1002988.

**Funding:** This work was supported by Centre National de la Recherche Scientifique (CNRS), Institut National de la Santé et de la Recherche Médicale (INSERM) and a grant from the CPER CTRL Longévité, Université de Lille (to MG). The funders had no role in study design, data collection and analysis, decision to publish, or preparation of the manuscript.

**Competing interests:** The authors have declared that no competing interests exist.

**Abbreviations:** AG, amylopectin granules; AID, auxin-inducible degron; APS, ammonium persulfate; DMSO, dimethyl sulfoxide; EM, electron microscopy; FDR, false discovery rate; HCD, higher-energy collisional dissociation; HFF, human foreskin fibroblast; IMC, inner membrane complex; MPA, mycophenolic acid; PAS, periodic acid–Schiff; PBS, phosphate-buffered saline; PM, plasma membrane; PFA, paraformaldehyde; TPP, trehalose-6-phosphate phosphatase; TPS, trehalose-6-phosphate synthase.

*Cryptosporidium* spp. (the cause of cryptosporidiosis). Interestingly, *T. gondii* has been used as a model for other apicomplexan parasites due to its genetic tractability. Acute toxoplasmosis is caused in the intermediate host by rapid proliferation of the tachyzoite form. Chronic toxoplasmosis is due to the establishment of the encysted form of the parasite (bradyzoites) in specific tissues (mainly brain and muscles). The establishment of the bradyzoite cysts in neurons or muscle cells is of interest in terms of pathology in humans, since reactivation of the cysts can lead to the deadly form of the disease (cerebral toxoplasmosis). It is also crucial for transmission since cysts contained in muscles are one of the main infectious routes for humans through the consumption of undercooked meat. Therefore, the ability to rapidly proliferate as a tachyzoite and to switch to the latent bradyzoite form are key elements of pathogenesis in humans and other intermediate hosts.

Diseases triggered by *T. gondii* involve an uncontrolled increase in parasite numbers, leading to inflammation and destruction of host cells. Although this parasite undergoes a sexual cycle in the definitive host, the pathogenesis is primarily driven by asexual replication cycles occurring within the host's cells. The tachyzoite proliferation involves a tightly regulated control of the cell cycle, ultimately resulting in the production of new daughter cells containing 1 nucleus and a complete set of organelles [1]. Although regulation of the cell cycle involves transcriptional control of gene expression [2,3], posttranslational regulation of critical mechanisms has been shown to occur in these parasites [1]. Among these posttranslational modifications, phosphorylation and dephosphorylation regulate important molecular functions including parasite cell division [4,5] and both *T. gondii*'s and *P. falciparum*'s global phosphoproteome show extensive phosphorylation of a large proportion of proteins [6] suggesting an important contribution of these posttranslational marks in the life cycle of these parasites. Moreover, phosphatases and kinases have been shown to play a crucial role during the *Plasmodium* life cycle [7,8].

*T. gondii* kinases have been demonstrated to have crucial roles in the division of the tachyzoite [9]. Among the 40 kinases that were inspected, 15 were essential to tachyzoite growth, many of which showed cell cycle defects linked to daughter cell formation and nuclear division [9]. Tachyzoite division is controlled by a bipartite centrosome, for which the inner core controls nuclear division while the outer core controls the daughter cell production [10]. The proper division of the centrosome is regulated by TgCDPK7 [11], TgMAPK-L1 [10], and TgMAPK2 [12]. Moreover, cell division of the *T. gondii* tachyzoite is regulated by 5 cyclically expressed kinases denoted as TgCrks; these kinases control several cell cycle checkpoints and ensure the smooth progression of the cell cycle [13,14]. Only a handful of protein phosphatases have been characterized in depth in this parasite and many protein phosphatase functions remain to be unraveled [5]. Seventeen serine/threonine protein phosphatases were mutated using the CRISPR/Cas9 system and only TgPP7 demonstrated a prominent role in virulence [15]. TgPP2A has been shown to play a role in differentiation from the tachyzoite to the bradyzoite stage by regulating starch metabolism [16,17], a process that is also controlled by the kinase TgCDPK2 [18]. TgPPKL depletion leads to the uncoupling of the cell cycle, with normal DNA duplication but severe defects in forming daughter parasites [19]. Moreover, TgPP1, a serine/threonine phosphatase with homologs that have been extensively studied in higher eukaryotes, has been shown to play a critical role in promoting motility of the parasite in response to $Ca^{2+}$ upon egress from the host cell [20]. Notably, upon the release of Ca2+ from intracellular stores, TgPP1 exhibited relocalization to the apex of the parasite [20]. Phosphoproteomics analysis after treating parasites with zaprinast (a cGMP activator that ultimately stimulates egress of the parasite), revealed that TgPP1 is likely involved in dephosphorylating crucial targets during egress and motility [20]. The affected pathways included transmembrane transport, cyclic nucleotide synthesis, ubiquitin transfer, and cytoskeletal motor activity [20],

revealing the crucial roles of TgPP1 in the control of the ultimate steps of the intracellular cycle of the parasite. The *P. falciparum* PP1 showed similar defects in egress of the host cell and also exhibited a deficiency in DNA replication [21]. *P. berghei* PP1 was shown to be crucial for the erythrocytic cycle [22] as well as for maturation of the male gametocyte and exflagellation, a process that requires multiple rounds of nuclear division [23]. The TgPP1 depleted parasites were also shown to exhibit growth defects independent from the egress phenotype but this was not explored further [20]. Therefore, the role of TgPP1 during the cell cycle and in controlling intracellular growth has not been investigated. Indeed, TgPP1, as well as *Plasmodium* PP1 [24–26], forms a complex with TgLRR1 [27] and TgI2 [28], 2 homologs of known regulators of cell cycle function in other eukaryotes [29], suggesting a role in controlling the cell cycle in this parasite.

In this study, we produced an inducible knockdown mutant of TgPP1 and characterized its phenotypes during the tachyzoite cell cycle. We demonstrate that TgPP1 has a crucial role in regulating the multiple pathways that are essential for the production of daughter cells such as inner membrane complex (IMC) assembly, organelle division, and chromosome segregation. TgPP1 also ensures normal assembly of the daughter cell's IMC. Phospho-proteomics demonstrated the differential phosphorylation of several IMC proteins and other targets that are linked to the regulation of the cell cycle. Moreover, we demonstrated the role of TgPP1 in starch production and identified an unknown protein that is crucial for the balance of starch production and whose activity is, at least partially, regulated through dephosphorylation by TgPP1.

## Materials and methods

### Parasite culture, transfection, and purification

*Toxoplasma gondii* tachyzoites belonging to the Type I RH Δ*ku80* Tir1 strain were grown in vitro in human foreskin fibroblast (HFF) cells using Dulbecco's Modified Eagles Medium supplemented with 10% fetal bovine serum (FCS), 2 mM glutamine, and 1% penicillin-streptomycin. The Type I Δ*ku80* Tir1strain has the *Ku80* gene deleted with the aim of promoting homologous recombination. In addition to expressing the Transport Inhibitor Response (Tir1) protein which has a role in the inducible degradation of targeted proteins using an auxin-inducible degradation (AID) system. Parasite strains were grown in T25 ventilated flasks containing HFF cell monolayers and maintained within a HERAcell VLOS 160i $CO_2$ incubator (Thermo Scientific) at 37°C and 5% $CO_2$. Electroporation was used to introduce transgenes into the parasite's genome by using a BTX Electro Cell Manipulator 600 at 1.5 kV.cm$^{-1}$, 25 μF capacitance, and 24 Ω resistance. Transgenic parasites were selected by using 25 μg/ml mycophenolic acid (MPA) and 50 μg/ml xanthine. Parasite clones were produced by using limiting dilution. Experiments consisting of DNA, total RNA, and protein purification involved tachyzoite collection by mechanical lysis using a 17-guage needle followed by a 24-guage needle (Terumo AGANI) followed by the filtration of the lysate using a 3 μm polycarbonate membrane (Whatman).

### Generation of transgenic *T. gondii* strains

The iKD TgPP1 line was generated by utilizing the Type I RH Δ*ku80* Tir1 strain as a parental strain. A CRISPR/Cas9 plasmid containing a gRNA targeting the 5′ end of the endogenous *TgPP1* gene (TGGT1_310700) and a PCR product consisting of the HXGPRT selection cassette (HXGPRT-T2A-AID-Ty) flanked by 30 bp of homology were used for transfection. The PCR product contained a Skip peptide (T2A) allowing for the dual expression of protein under a single promoter followed by cleavage of the TgPP1 protein from the HXGPRT protein

in order to promote the degradation of the TgPP1 protein by the proteasome once the auxin hormone is added. The PCR product was amplified by using the pHXGPRT- 2TA-AID-Ty plasmid as a template. Ten μg of PCR product was transfected with 30 μg of pSAG1:: Cas9-U6 targeting the 5′ end of the TgPP1 gene in the RH Δ*ku80* Tir1 strain. Primers used to generate transfection material are included in S1 Table.

### Generation of trehalose phosphatase mutants

TGGT1_297720 gene was tagged at the 3′ end with 3xmyc tag in the RH Δ*ku80* strain using a CrispR/Cas9 expressing construct targeting the 3′ end of the gene. To generate point mutation in the TGGT1_297720 gene, a CrispR/Cas9 construct was designed in the 15th intron of the gene. A DNA fragment that included the 2 phosphosites to be mutated was synthesized with a point mutation in the PAM site of the gRNA (located in the 15th intron of the gene). The mutated DNA fragments were generated by mutagenesis using the Q5 mutagenesis kit to induce point mutations in the codons corresponding to Serine 1054 and Serine 1073. They were mutated to either Alanine or Aspartate amino acids. These DNA fragments were co-transfected together with the CrispR/Cas9 construct into the RH Δ*ku80* strain. FACS sorting based on the CrispR/Cas9 construct's GFP expression was used to isolate the mutant strains and directly clone them into 96-well plates. Each mutant clone was verified using sequencing.

### Growth assays

Growth assays were carried out by the inoculation of $8 \times 10^4$ parasites of parental Tir1 and iKD TgPP1 mutant parasites on HFF cell monolayers grown on coverslips in a 24-well plate for 24 h and 48 h in the presence and absence of 0.5 mM of auxin (AID/indoleacetic acid). Auxin was introduced into the media with the aim of inducing TgPP1 protein degradation. After either 24 h or 48 h, infected coverslips were fixated with 4% of paraformaldehyde (PFA). The fixated parasites were stained using anti-TgEno2 in order to stain parasite nuclei as well as anti-TgIMC1 antibodies in order to stain the IMC. The number of parasites per vacuole was counted for a total of 100 vacuoles per biological replicate. Each growth assay experiment included a total of 3 to 5 biological replicates.

### Plaque assays

In order to carry out the plaque assays, a total of 1,000 parasites of either the Parental Tir1 strain or iKD TgPP1 mutant strain were inoculated on a monolayer of HFF cells grown in a 6-well plate either in normal media or media treated with 0.5 mM of auxin. Parasites were left to grow for 7 days before fixation using 100% ethanol. Plaques were stained using Crystal Violet for the purpose of plaque visualization. An Excel macro was used to quantify the plaque size for each experimental condition.

### Organelle labeling

Parental Tir1 and iKD TgPP1 parasites were left to grow on monolayers of HFF cells grown on coverslips of 24-well plates in normal media as well as media treated with auxin for 24 h and 48 h before fixating parasites with 4% PFA and staining with antibodies. The IMC was labeled using anti-TgIMC1, anti-TgISP1, and anti-TgGAP45. Other organelles consisting of the Golgi, plastid, and centrosome were labeled using anti-TgSORT, anti-TgACP, and anti-TgCentrin1 antibodies, respectively.

## Immunofluorescence assays

Immunofluorescence experiments were carried out after fixation of intracellular parasites grown on coverslips using 4% of PFA for 30 min. This was followed by coverslip washing 3 times using 1× PBS buffer. Permeabilization was carried out for 30 min using the following buffer (1× PBS, 0.1% Triton 100×, 0.1% glycine, 5% FBS). This was followed by primary antibody incubation for 1 h. Antibodies were diluted using IFA buffer used for permeabilization. Coverslips containing fixed intracellular parasites were then washed 3 times using 1× PBS. Coverslips were then incubated for 1 h in DAPI and secondary antibodies coupled to either Alexa-594 or Alexa-488. Coverslips were then washed 3 times using 1× PBS buffer and mounted onto microscope slides using Moviol. Primary antibodies used included anti-TgIMC1 (a gift from Prof. Ward, University of Vermont), anti-TgEno2, anti-TgISP1 [30], anti-TgCentrin1 (a gift from Prof. Gubbels, College of Boston), anti-TgACP (a gift from Pr. Striepen, U. Penn), anti-TgSortilin, and anti-TgGAP45 (Prof. Soldati, Geneve University), anti-HA, and anti-Ty (a gift from Dr. Bastin, Institut Pasteur de Paris) were used at the following dilutions: 1:500, 1:1,000, 1:500, 1:500, 1:500, 1:500, 1:10,000, 1:500, and 1:10,000, respectively. Signal was visualized manually by counting 100 parasites for each replicate. A total of 3 replicates were carried out for each experiment.

Immunofluorescence assay experiments were visualized using the ZEISS LSM88O confocal microscope at a magnification of 63×. Image processing was carried out using the CARL Zeiss Zen software.

## Electron microscopy and cytochemical localization of glycogen

iKD TgPP1 parasites were grown on monolayers of HFF cells in normal media or media treated with auxin for 48 h in T25 flasks. Intracellular parasites were fixated using the following solution: 1% glutaraldehyde in 0.1 M sodium cacodylate at a pH of 6.8 and at 4°C. Parasite samples were then post-fixated using 1% osmium tetraoxide and 1.5% potassium ferricyanide. This was then followed by using 1% uranyl acetate. Post-fixation was carried out for 1 h using the following conditions: distilled water, in the dark, and at room temperature. Increasing ethanol concentration solutions were used to dehydrate fixated samples after washing. Epoxy resin was used with the aim of infiltrating parasite samples. This was followed by curation for 24 h at 60°C. Deposition of 70 to 80 nm-thick sections was carried out in formvar-coated grids. Images were observed using 80 kV on a Hitachi H7500 TEM (Federal University of Rio de Janeiro University, Brazil). Acquisition of images was carried out by using an Mpixel digital camera (Federal University of Rio de Janeiro University, Brazil). For the cytochemical localization of glycogen, parasites were fixed and processed as described for TEM. Subsequently, 90-nm sections were collected on gold grids and incubated for 15 min in a solution containing 1% periodic acid, washed 3 times, and then incubated with 1% thiosemicarbazide in 10% acetic acid for 24 h. Successive washes were performed in 10%, 5%, and 2% acetic acid for 10 min each. Afterward, the sections were incubated with 1% silver proteinate for 30 min, protected from light. Finally, the sections were washed in distilled water for 10 min each and observed unstained on a FEI Tecnai SPIRIT transmission electron microscope.

## Ultrastructure expansion microscopy

Coverslips with HFF monolayers were inoculated with the iKD TgPP1 or ikD TgPP1-IMC17-myc parasite strains. They were cultured either in normal media or media treated with auxin for 24 h. Following this, cells were fixed using 4% PFA for 20 to 30 min and prepared for ultrastructure expansion microscopy (U-ExM). In brief, coverslips were incubated for 5 h at 37°C in a solution of 2× 1.4% acrylamide (AA) and 2% formaldehyde (FA). Gelation was achieved

by incubating the samples in a mixture containing ammonium persulfate (APS), tetramethyl-lethylenediamine (TEMED), and a monomer solution (19% sodium acrylate, 10% AA, and 0.1% bis-acrylamide in 10× PBS) for 1 h at 37˚C. Post-gelation, the gels were denatured at 95˚C for 1.5 h. The denatured gels were then expanded by incubating overnight in double-distilled H2O (ddH2O). The following day, gels were washed 3 times with PBS (10 min each wash) before incubating with primary antibodies for 3 h at 37˚C. After primary antibody incubation, the gels were washed 3 times in PBS-Tween 0.1%, followed by an overnight incubation with secondary antibodies at 4˚C. Subsequent washes in PBS-Tween 0.1% were performed 3 times before a second round of expansion in ddH2O prior to imaging. Confocal imaging was performed using a ZEISS LSM880 Confocal Microscope with a 63× magnification. Primary antibodies used were anti-TgIMC3 (provided by Prof. Gubbels, Boston College) at a dilution of 1:1,000, acetylated α-tubulin (Santa Cruz Biotechnology) at a dilution of 1:200, anti-Myc (abcam) at a dilution of 1:200, and anti-TgNuf2 (provided by Prof. Gubbels, Boston College) used at a dilution of 1:200.

## RNA sample preparation and extraction

RNA samples of iKD TgPP1 parasites were collected after inoculating parasite onto monolayers of HFF cells grown in T175 flasks. iKD TgPP1 parasites were grown in normal media (control) as well as media treated with auxin for 24 h. RNA extraction was carried out by re-suspending the parasite sample in Trizol (Invitrogen). This was followed by the addition of chloroform (4˚C) to the sample allowing for the separation of the RNA-containing aqueous phase and the protein-containing organic phase. This was followed by a centrifugation step at room temperature for 10 min. The aqueous phase was then transferred into a new tube consisting of cold isopropanol in order to precipitate the RNA followed by a centrifugation step. Washing of the precipitated RNA pellet was carried out using 70% ethanol followed by air-drying the pellet and re-suspension in RNase-free water (Gibco). RNA samples were purified using the RNase-free DNaseI Amplification Grade Kit (Sigma). Extracted RNA quality was verified using an RNA 6000 Nano (Agilent) chip and RNA samples with an integrity score of 8 or greater were used for RNA library preparation.

## RNA-sequencing library preparation

Libraries for the RNA samples were prepared by using the TruSeq Stranded mRNA Sample Preparation Kit (Illumina). Libraries were prepared as per the manufacturer's instructions. Validation of the libraries was carried out using DNA high sensitivity chips read by an Agilent 2100 Bioanalyzer. Libraries were then quantified by using the KAPA library quantification kit (Illumina) using a 12K QuantStudio qPCR thermocycler.

## RNA-sequencing analysis

RNA libraries were sequenced using a HiSeq 2500 as 50 bp reads by using the sequence by synthesis technique. HiSeq control software and real-time analysis component were used for image analysis. Bcl2fastq 2.17 (Illumina) was used to demultiplex. Data set quality was verified using FastQC v0.11.8–0. Cutadapt v1.18. was used to treat the adapters for sequencing. Filtering of reads shorter than 50 bp and low-quality bases was carried out by using Trimmomatic v0.38.1. Alignment of data sets using HiSAT2 v2.2.0 was carried out after cleaning of data sets against the *T. gondii* ME49 genome from ToxoDB. Annotated gene expression was quantified using htseq-count from the HTseq suite v0.9.1. DeSeq2 v1.22.1 was used to carry out differential gene expression analysis. *P*-values were adjusted using the Benjamin–Hochberg method.

Adjusted *p*-values less than 0.05 and a log2 fold change greater than 2 corresponding to differentially expressed genes were kept.

## Phosphoproteomics sample extraction

Mutant iKD TgPP1 parasites were left to grow in T175 flasks before the addition of auxin for 2 h and 24 h in T175 flasks. iKD TgPP1 parasites grown in normal media were considered as a control. After parasite purification by filtration and centrifugation, parasite pellet was re-suspended using 8 M Urea consisting of Protease and Phosphatase Inhibitor Cocktail (Thermo Fisher Scientific). Parasite was re-suspended in urea solution to a final concentration of 35 million parasites/50 µl of re-suspension solution. Extraction steps were carried out on ice. A Biorupter Sonicator was used at 4°C for 10 cycles (30 s on/off per cycle). This was followed by a centrifugation step for 20 min at 4,000 rpm at 4°C. The supernatant was separated from the pellet and put into a new tube. The Pierce BCA Protein Assay (Life Technologies) was used to quantify the protein concentration of the samples. Samples were then stored at −80°C prior to analysis.

## Tryptic digestion

S-Trap mini spin column (Protifi, Hutington, United States of America) digestion was performed on 100 µg of cell lysates, according to manufacturer's instructions. Briefly, SDS concentration was first adjusted to 5% and samples were reduced with 20 mM TCEP and alkylated with 50 mM chloracetamide for 15 min at room temperature. Aqueous phosphoric acid was then added to a final concentration of 1.2% following by the addition of S-Trap binding buffer (90% aqueous methanol, 100 mM TEAB, pH 7.1). Mixtures were loaded on S-Trap columns by 30 s centrifugation at 4,000 × g; 6 washes were performed before adding the trypsin (Promega) at 1/20 ratio for 2 h at 47°C. After elution, peptides were vacuum dried.

## Phosphopeptides enrichment by titanium dioxide (TiO₂) and phosphopeptides purification by graphite carbon (GC)

Phosphopeptide enrichment was carried out using a Titansphere $TiO_2$ Spin tip (3 mg/200 µl, Titansphere PHOS-TiO, GL Sciences, Japan) on 90 µg of digested proteins for each biological replicate. Briefly, the $TiO_2$ Spin tips were conditioned with 20 µl of solution A (80% acetonitrile, 0,1% TFA), centrifuged at 3,000 × g for 2 min and equilibrated with 20 µl of solution B (75% acetonitrile, 0,075% TFA, 25% lactic acid) followed by centrifugation at 3,000 × g for 2 min. Peptides were resuspended in 20 µl of 10% acetonitrile, 2% TFA in HPLC-grade water, mixed with 100 µl of solution B and centrifuged at 1,000 × g for 10 min. Sample was applied back to the TiO2 Spin tips 2 more times in order to increase the adsorption of the phosphopeptides to the TiO2. Spin tips were washed with, sequentially, 20 µl of solution B and 2 times with 20 µl of solution A. Phosphopeptides were eluted by the sequential addition of 50 µl of 5% $NH_4OH$ and 50 µl of 5% pyrrolidine. Centrifugation was carried out at 1,000 × g for 5 min.

Phosphopeptides were further purified using GC Spin tips (GL-Tip, Titansphere, GL Sciences, Japan). Briefly, the GC Spin tips were conditioned with 20 µl of solution A, centrifuged at 3,000 × g for 2 min and equilibrated with 20 µl of solution C (0,1% TFA in HPLC-grade water) followed by centrifugation at 3,000 × g for 2 min. Eluted phosphopeptides from the TiO2 Spin tips were added to the GC Spin tips and centrifuged at 1,000 × g for 5 min. GC Spin tips were washed with 20 µl of solution C. Phosphopeptides were eluted with 70 µl of solution A (1,000 × g for 3 min) and vacuum dried.

## nanoLC-MS/MS protein identification and quantification

Samples were resuspended in 42 μl of 0.1% TFA in HPLC-grade water. For each run, 5 μl was injected in a nanoRSLC-Q Exactive PLUS (RSLC Ultimate 3000, Thermo Scientific, Massachusetts, USA). Phosphopeptides were loaded onto a μ-precolumn (Acclaim PepMap 100 C18, cartridge, 300 μm i.d. × 5 mm, 5 μm, Thermo Scientific, Massachusetts, USA) and were separated on a 50 cm reversed-phase liquid chromatographic column (0.075 mm ID, Acclaim PepMap 100, C18, 2 μm, Thermo Scientific, Massachusetts, USA). Chromatography solvents were (A) 0.1% formic acid in water; and (B) 80% acetonitrile, 0.08% formic acid. Phosphopeptides were eluted from the column with the following gradient 1% to 40% B (120 min), 40% to 80% (1 min). At 121 min, the gradient stayed at 80% for 5 min and, at 126 min, it returned to 5% to re-equilibrate the column for 20 min before the next injection. Two blanks were run between each replicate to prevent sample carryover. Phosphopeptides eluting from the column were analyzed by data-dependent MS/MS, using top-10 acquisition method. Phosphopeptides were fragmented using higher-energy collisional dissociation (HCD). Briefly, the instrument settings were as follows: resolution was set to 70,000 for MS scans and 17,500 for the data-dependent MS/MS scans in order to increase speed. The MS AGC target was set to $3.10^6$ counts with maximum injection time set to 200 ms, while MS/MS AGC target was set to $1.10^5$ with maximum injection time set to 120 ms. The MS scan range was from 400 to 2,000 m/z. Dynamic exclusion was set to 30 s duration.

For total proteomic analysis, peptides were eluted from the column with the following gradient 5% to 40% B (120 min), 40% to 80% (6 min). At 127 min, the gradient returned to 5% to re-equilibrate the column for 20 min before the next injection. MS parameters were the same as those used for LC-MS/MS analysis described above.

## Data processing following nanoLC-MS/MS acquisition

The MS files were processed with the MaxQuant software version 1.6.14.0 and searched with the Andromeda search engine against the UniProtKB/Swiss-Prot *Homo sapiens* database (release February 2021, 20,396 entries) and *Toxoplasma gondii* strain ATCC 50853/GT1 (release November 2020, 8,450 entries). To search parent mass and fragment ions, we set an initial mass deviation of 4.5 ppm and 0.5 Da, respectively. The minimum peptide length was set to 7 amino acids and strict specificity for trypsin cleavage was required, allowing up to 2 missed cleavage sites. Carbamidomethylation (Cys) was set as fixed modification, whereas oxidation (Met), N-term acetylation, and phosphorylation (Ser, Thr, Tyr) were set as variable modifications (only for phosphoproteomics analysis). The match between runs option was enabled with a match time window of 0.7 min and an alignment time window of 20 min. The false discovery rates (FDRs) at the protein and peptide level were set to 1%. Scores were calculated in MaxQuant as described previously [31]. The reverse and common contaminants hits were removed from MaxQuant output.

The phosphopeptide output table and the corresponding logarithmic intensities were used for phosphopeptide analysis. The phosphopeptide table was expanded to separate individual phosphosites, and all sites identified in all 4 replicates were kept in at least 1 group for the 2 h condition and in all 3 replicates in at least 1 group for the 24 h experiment. Missing values were inputed using width = 0.3 and down-shift = 3. Significantly altered phosphosites were represented by volcano plots (*t* test S0 = 1, FDR = 0.05).

For total proteome analysis, only proteins identified in all 4 replicates were kept in at least 1 group. Missing values were inputed using width = 0.3 and down-shift = 2.5. The significantly altered proteins were represented on a volcano plot s (*t* test S0 = 1, FDR = 0.05).

## Western blotting

Western blotting was carried out by inoculating $2 \times 10^6$ parasites of the iKD TgPP1 strain in T175 flasks grown in normal media for 24 h (control) and grown in media treated with auxin for 1 and 2 h. Parasite samples were collected by filtration, followed by centrifugation. The obtained pellet was re-suspended in loading buffer consisting of 240 mM Tris-HCl (pH 6.8), 8% SDS, 40% saccharose, 0.04% bromophenol blue, and 400 mM DTT. This was followed by a denaturing step by incubating the parasite samples at 95˚C for 10 min. Protein extracts were separated by electrophoresis on an 8% polyacrylamide gel and then transferred onto a nitrocellulose membrane (GE Healthcare) for 90 min at 100 V. Blocking buffer containing 5% milk in TNT buffer consisting of 100 mM Tris (pH 8), 150 mM NaCl, and 0.1% Tween was used to block the membrane. Western blot membranes were incubated in primary anti-body for 1 h, washed 4 times, incubated for another hour in the secondary antibody. Super Signal West Femto Maximum Sensitivity Substrate (Thermo Scientific) was used to reveal protein bands and ChemiDoc XRS$^+$ (Biorad) was used to visualize protein bands. Antibodies used were anti-Ty used at a dilution of 1/500. Secondary antibody used is species specific and conjugated to HRP.

## PAS staining

For the PAS staining, Parental Tir 1 and iKD TgPP1 parasites were left to grow on HFF cell monolayers grown on coverslips for 48 h in the presence and absence of auxin. Periodic acid-Schiff stain was used in order to determine the presence of polysaccharides within the parasites. Visualization of polysaccharides was carried out by using a confocal microscope.

## Amylopectin quantification

Parental Tir1 and iKD TgPP1 parasites were left to grow for 48 h in the presence and absence of auxin before parasite filtration and centrifugation. Around 200 million parasites were collected for each sample. After purification, parasite samples were stored at −80˚C and sent for analysis. Purified *T. gondii* cells were resuspended in ice-cold phosphate-buffered saline (PBS) at a concentration of 20 million parasites per ml. Cell suspensions were then disrupted 3 times by a French press (13,000 p.s.i.) and centrifuged at 10,000g for 30 min at 4˚C. The pellets containing amylopectin and cell debris were passed through a self-formed 90% Percoll gradient at 10,000g for 30 min at 4˚C. The purified amylopectin pellets were washed in ultrapure water, centrifuged twice at 10,000g, and kept dry at 4˚C. Polysaccharide amounts were measured by an amyloglucosidase assay using the Enzytec starch kit following the manufacturer's recommendations. In parallel, 30 μg of polysaccharide were resuspended in 20 μl of 100% dimethyl sulfoxide (DMSO), boiled 10 min, and diluted to 10% DMSO. Twenty μl of a freshly prepared iodine solution (0,2% I2; 2% KI) were added to 80 μl of the boiled sample and the absorbance of the complex was monitored from 700 to 400 nm allowing the determination of its λmax (wavelength at the maximal absorbance). Controls including commercial potato amylopectin, potato amylose, and rabbit liver glycogen (Sigma) were used and displayed λmax values of 542 nm, 631 nm, and 495 nm, respectively.

## Statistics

Graph pad Prism software version 8/9 (San Diego, California, USA) was used to analyze all data concerning growth assays, proliferation assays, and ratio counts. Student *t* tests were used to determine significant differences between data sets where *p*-values <0.05 were considered

as significant. All experiments were carried out in biological triplicates. For each independent experiment, a total of 100 parasites/vacuoles was counted.

# Results

## TgPP1 is essential for the normal growth and proliferation of *T. gondii* tachyzoites

We generated an inducible knock down mutant of TgPP1(iKD TgPP1) with the aim of determining the role of the TgPP1 protein using the auxin-inducible degron (AID) system. The mutant parasite line was produced using CRISPR/Cas9 to insert an *hxgprt-t2a-AID-2ty* insert at the 5′ end of the TgPP1 gene (Fig 1A). Previous attempts to generate the transgenic parasite strain by inserting an *hxprt-AID-HA* cassette at the 3′ end were unsuccessful. Insertion of the *hxgprt-t2a-AID-2xty* cassette at the appropriate locus within the iKD TgPP1 genome was confirmed by insertion PCR (S1A Fig). The localization of Ty-tagged TgPP1 protein was investigated using IFA and TgPP1 was demonstrated to be localized mainly in the nucleus but also in the cytoplasm when compared with a cytoplasmic marker (TgAlba1 [32]) during the tachyzoite intracellular cell cycle (Fig 1B). The depletion of TgPP1 was obtained within 1 h of auxin treatment as verified by western blot of total protein extracts (Fig 1C). A growth assay was carried out for 24 h in the presence and absence of auxin. We noticed that the growth of the strain

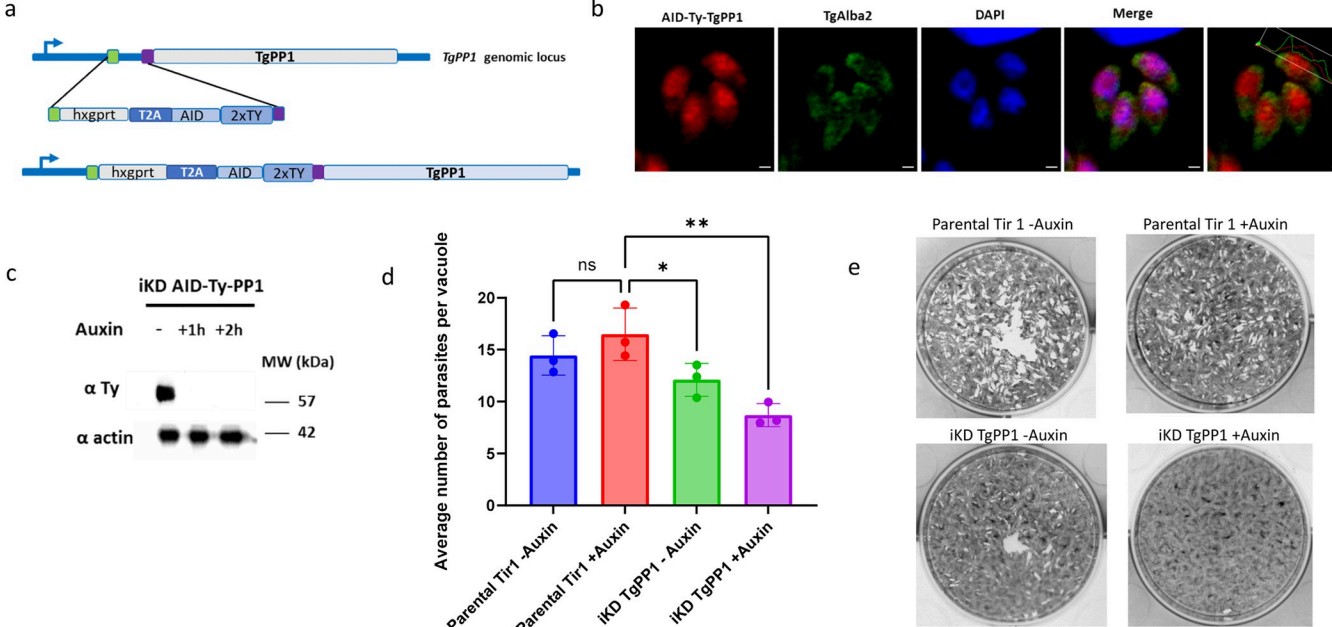

**Fig 1. TgPP1 is required for parasite growth and proliferation. (a)** Schematic representation of the construct used to generate iKD TgPP1 mutant parasites using the AID system that allows for the inducible degradation of the TgPP1 protein. The system involves introducing the AID domain into the gene of interest. In the presence of Auxin, the AID domain will be recognized by the Tir1 protein and the protein degraded by the proteasome. The HXGPRT-T2A-AID-2Ty cassette is inserted at the 5′ end of the gene using the CRISPR/Cas9 gene-editing strategy. **(b)** Confocal microscopy imaging of the AID-Ty iKD TgPP1 parasite in the absence of auxin labeled with anti-Ty (red, TgPP1) and TgAlba2 (green), a protein that localized to the cytoplasm. DAPI (blue) was used to stain the nucleus. The scale bar is indicated in the lower right corner of each image. Measurements of red and green fluorescence throughout length of the parasite is indicated in the upper right corner of the overlayed TgPP1 and TgAlba2 image. **(c)** Western blot of total protein extract from the iKD TgPP1strain in absence and presence of auxin (1 h and 2 h) displaying depletion of the TgPP1 protein after the addition of Auxin. Western blots were probed with anti-Ty antibodies to determine the presence of the TgPP1 protein (upper panel). Western blot was normalized using anti-TgActin antibodies (lower panel). **(d)** Growth assay of the Parental Tir1 and iKD TgPP1 strains in the absence and presence of Auxin for 48 h. The average number of parasites per vacuole was recorded. A Student's *t* test was performed. *$p < 0.05$, **$p < 0.01$; mean ± SD ($n = 3$). **(e)** Plaque assay demonstrating proliferation of the Parental Tir 1 and iKD TgPP1 strain in the presence and absence of auxin. The data underlying this figure can be found in S1 Data. AID, auxin-inducible degron.

was affected in absence of auxin, a phenotype that was also observed for another published TgPP1 conditional mutant [20] (S1B Fig). However, the iKD TgPP1 mutant in presence of auxin demonstrated a significant growth defect after 48 h (Fig 1D). We then performed a plaque assay that measures the ability of the parasite to form plaques in the host cell monolayer over a period of 7 days. In this assay, the iKD TgPP1 was unable to form plaques in the presence of auxin thus demonstrating that TgPP1 is important for parasite proliferation as a tachyzoite (Fig 1E). This experiment also showed that the size of the plaque produced by the mutant in absence of auxin was smaller than the parental parasite strain, confirming that the insertion of the AID construct was deleterious to the parasite's growth (S1C Fig). Overall, these results suggest the essentiality of TgPP1 for the normal growth and proliferation of the tachyzoite.

## TgPP1 is important for the formation of the inner membrane complex (IMC)

To better understand the biological role of TgPP1 during the intracellular tachyzoite cell cycle, we investigated its ability to form the IMC, a double membrane structure located below the tachyzoite's plasma membrane (PM). Upon depletion of TgPP1, the parasites exhibited an abnormal IMC structure as labeled by an anti-IMC3 antibody, featuring an unstructured network of intertwined IMC3 protein. Upon auxin treatment, the IMC was unable to form properly (Fig 2A, lowest panel) as opposed to the IMC structures of the iKD TgPP1 mutant in the absence of auxin treatment as well as the IMC structures of the Parental Tir1 strain (Fig 2A, upper and medium panels). We also performed expansion microscopy to better asses this phenotype (Fig 2B), showing parasites with similar defect of IMC formation. The quantification of the percentage of vacuoles with this IMC phenotype demonstrated that 40% of the vacuoles showed a similar phenotype after 48 h of auxin treatment (Fig 2C). This phenotype was accompanied by the presence of the parasite's IMC without nuclear material or nuclear material without formed IMC reminiscent of a missegregation of nuclear material as illustrated in Fig 2A (arrowed, lower panel, iKD TgPP1 + Auxin) or Fig 2B (arrowed, lower panel, iKD TgPP1 + Auxin). Quantification revealed that a similar percentage (40%) of vacuoles showed an important missegregation of nuclear material in the iKD TgPP1 mutant in the presence of auxin (Fig 2D) while this phenotype was almost lacking in the parental strain and in absence of auxin. The effect of TgPP1 depletion on additional components of the IMC was also investigated. Upon auxin treatment, TgGAP45, a protein targeted to the PM and connected with the IMC at the C-terminal end, revealed similar defects in IMC formation in the iKD TgPP1 mutant parasites treated with auxin (S2A Fig). Similar results were obtained with the IMC Sub-compartment protein 1 (TgISP1) which in normal instances localizes to the apical cap of the IMC and was instead dispersed along the periphery of the iKD TgPP1 parasite in the presence of auxin (S2B Fig). TgIMC1 was also used as an IMC marker and showed essentially similar phenotypes to TgIMC3 (S2C Fig, quantified in S2D Fig). To further confirm the effect of TgPP1 depletion on the IMC structure, electron microscopy (EM) was carried out. In absence of auxin, the iKD TgPP1 mutant parasites showed both the PM and the IMC normally formed and these 2 structures appeared intact (Fig 2D). In presence of auxin, the iKD TgPP1 mutant parasites exhibited an intact and continuous PM along the parasite. In contrast, the IMC was only partially present and discontinuous along the formed PM (black arrows, Fig 2E). Quantification of these EM observations (S2E Fig) further validate the role of TgPP1 in correctly forming the IMC.

## The effect of TgPP1 depletion affects the division and segregation of the Golgi and plastid

During the tachyzoite's cell cycle, the subcellular organelles are replicated according to a well-determined timeline [2]. Indeed, the centrosome divides first and is then followed by the Golgi

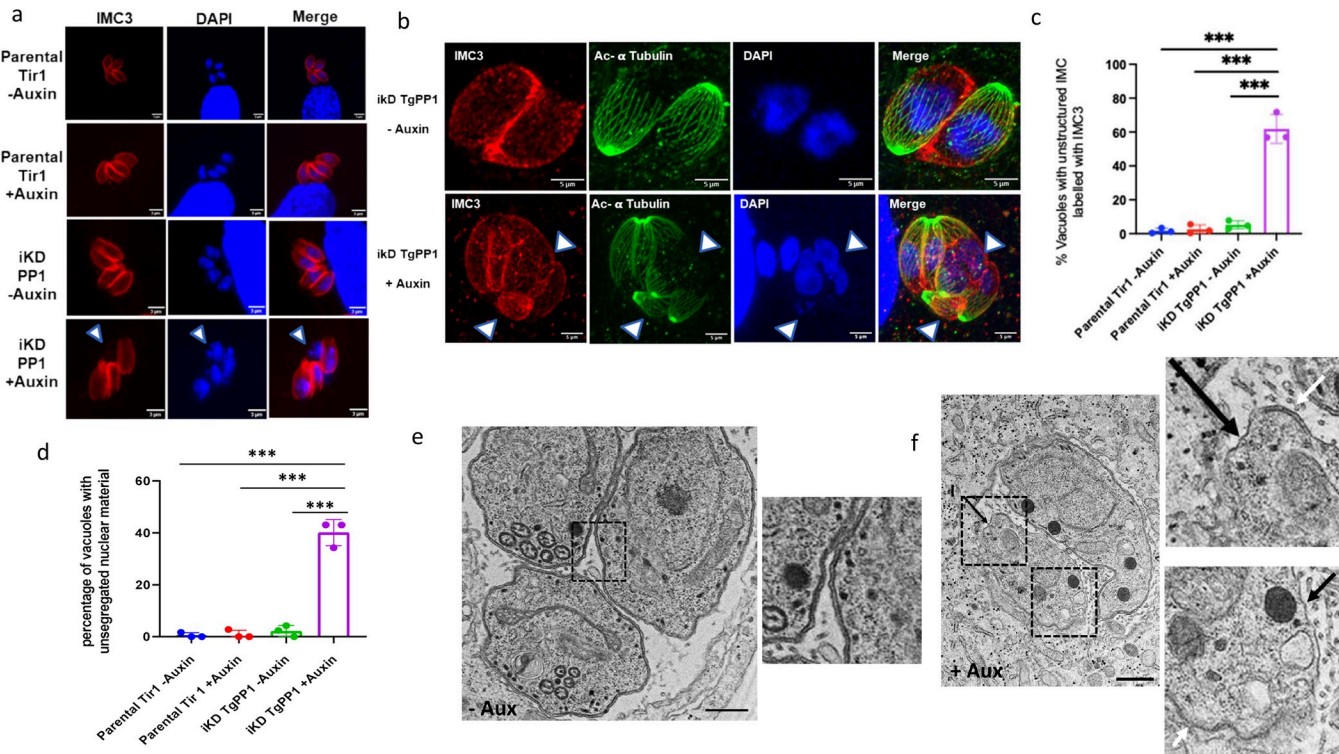

**Fig 2. Depletion of TgPP1 results in a collapsed IMC and unsegregated nuclei. (a)** Confocal imaging of the Parental Tir1 and iKD TgPP1 strains labeled TgIMC3 (red) in the presence and absence of auxin treatment. DAPI was used to stain the nucleus. Scale bar (1 μm) is indicated in the lower right corner of each individual image. A nucleus without formed parasite body is indicated by a white arrow. **(b)** Expansion microscopy images of the iKD TgPP1 strain in absence and presence of auxin. The parasite IMC (IMC3, red) and cytoskeleton (acetylated tubulin, green) were labeled as well as the nucleus by DAPI (blue). A parasite without nucleus and a parasite bearing 2 nuclei are indicated by a white arrow. **(c)** Bar graph representing the percentage of Parental Tir1 and iKD TgPP1 vacuoles possessing a collapsed IMC by using anti-TgIMC3 antibodies for labeling the IMC in the absence and presence of auxin treatment for 48 h. A Student's *t* test was carried out, \*\*\**p* < 0.001; mean ± SD (*n* = 3). **(d)** Bar graph displaying the quantification of vacuole with nuclear segregation defects in the Parental Tir1 and iKD TgPP1 strain in the absence and presence of auxin treatment for 48 h. A Student's *t* test was carried out, \*\*\**p* < 0.001; mean ± SD (*n* = 3). **(e)** EM image demonstrating the structural morphology characteristics of the IMC and PM of the iKD TgPP1 mutant parasite in the absence of auxin treatment. In this case, both IMC and PM remain intact Magnified region is boxed. **(f)** EM image demonstrating the structural morphology of the IMC and PM of the iKD TgPP1 parasite after auxin treatment for 48 h. In this case, PM membrane remains intact, but the IMC is absent. (I) stand for IMC. Scale bar (5 μm) is demonstrated in the lower right region of each EM. Magnified regions are boxed. Region missing the IMC are indicated by black arrows. A region with both the IMC and PM is indicated by a white arrow. The data underlying this figure can be found in S1 Data. EM, electron microscopy; IMC, inner membrane complex; PM, plasma membrane.

and then plastid [33]. We investigated the effect of TgPP1 depletion on the division of the centrosome (using TgCentrin1 as a marker of the outer core centrosome) by IFA. However, when counting more than 100 vacuoles in each of the 3 biological replicates, no significant change of the TgCentrin1 to nucleus ratio was identified, suggesting that the centrosome division is mainly unaffected in absence of TgPP1. We then investigated the effect of TgPP1 on the division of organelles. We, therefore, studied the effect of TgPP1 on the division of the plastid and the Golgi. By IFA, we observed that the plastid (TgCpn60) and Golgi (TgSORT) were missegregated after TgPP1 depletion (Fig 3A). Representative pictures presenting the normal and abnormal phenotype observed are presented in Fig 3B. The depletion of TgPP1 after 48 h of auxin treatment resulted in a significant percentage of parasite vacuoles with this phenotype (Fig 3C). To better assess the role of TgPP1 during division and cell cycle, we performed expansion microscopy (S3C Fig) and assessed the number of parasites undergoing mitosis (S3D Fig). While parasites were undergoing mitosis at a similar rate in auxin treated and control samples, the number of parasites in metaphase was much higher in the auxin treated

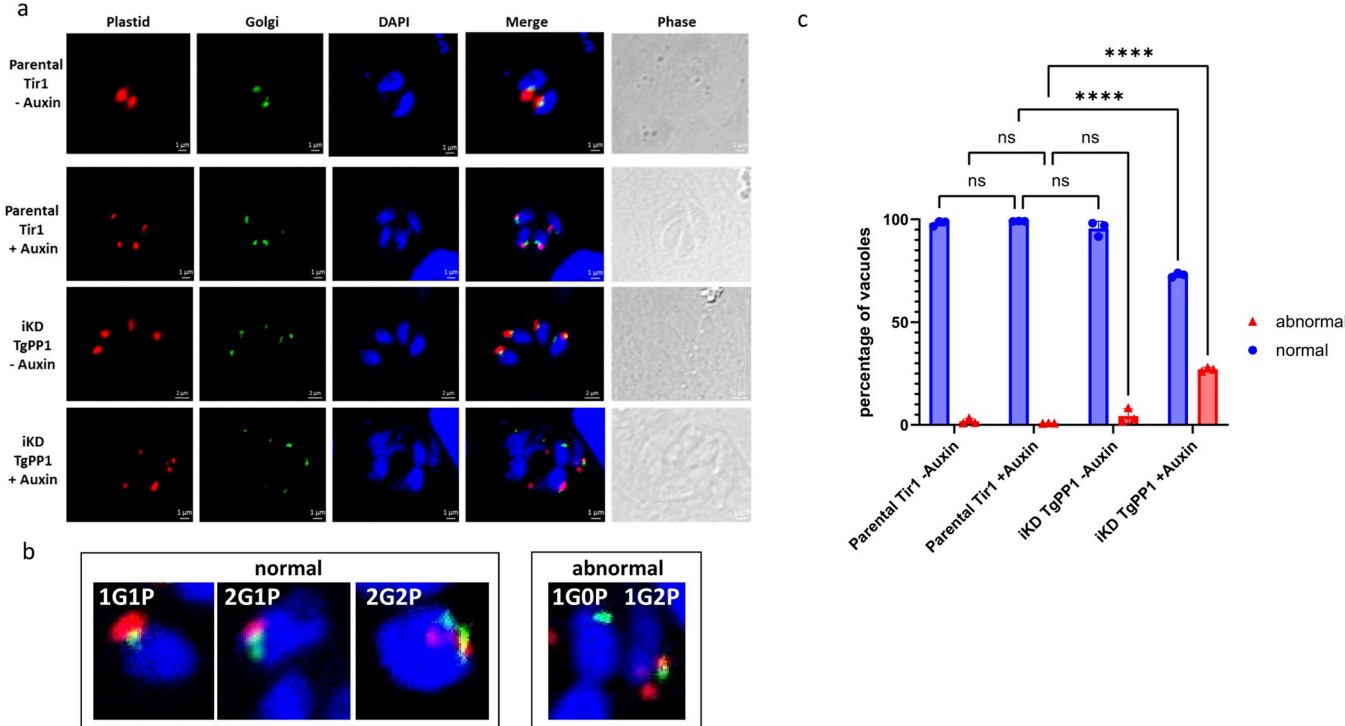

**Fig 3. TgPP1 depletion induce plastid and Golgi missegregation. (a)** Confocal imaging of the Parental Tir1 and iKD TgPP1 strains with labeled plastid (red) and Golgi (green) in the presence and absence of auxin treatment. DAPI was used to stain the nucleus. Scale bar (1 μm) is indicated in the lower right corner of each individual image. **(b)** Zoom images representing the normal segregation of Golgi (G, green) and plastid (P, red). A zoom image representing the abnormal segregation of Golgi (G, green) and plastid (P, red) is also shown on the right panel. DAPI was used to stain the nucleus. **(c)** Graph bar demonstrating the percentage of Parental Tir1 and iKD TgPP1 vacuoles possessing normal and abnormal plastid and Golgi segregation in the absence and presence of auxin treatment for 48 h. A Student's *t* test was carried out. ****$p < 0.0001$; mean ± SD ($n = 3$). Blue bars represent normal plastid and Golgi segregation. Red bars represent abnormal plastid and Golgi segregation. The data underlying this figure can be found in S1 Data.

parasites, indicating that these parasites may be slowed down at this stage after depletion of TgPP1. Collectively, these results demonstrate that TgPP1 has an important role in coordinating the cell cycle steps after centrosome division.

## TgPP1 depletion affects amylopectin steady-state levels

The control of amylopectin levels has been reported to be regulated by the phosphorylation [18] and dephosphorylation [16,17] of enzymes of amylopectin metabolism in *T. gondii*. We hypothesized that TgPP1 could also play a role in starch storage. To measure the effect of TgPP1 depletion on the amylopectin steady-state levels, we used the periodic acid–Schiff (PAS) method that labels polysaccharides. After 48 h of TgPP1 depletion, we clearly identified the appearance of structures positive for PAS staining accumulating in the parasites (Fig 4A, lower panel). This labeling is almost inexistent in the parental strain (in presence or absence of auxin) but can be detected in the iKD TgPP1 mutant in absence of auxin (Fig 4A). The presence of amylopectin granules was further confirmed through EM using a similar staining. Amylopectin granules (AG) were rare in the iKD TgPP1 parasites in the absence of auxin (Fig 4Bi) but observed in the iKD TgPP1 parasites in the presence of auxin at 24 h and 48 h (Fig 4Bii and 4Biii, respectively). AG accumulated in the cytoplasm of the parasite but not in the residual body, a phenotype that is different from what is observed in absence of TgCDPK2 or TgPP2A. The percentage of PAS positive vacuoles was quantified in the iKD TgPP1 mutant and parental strains in the absence and presence of

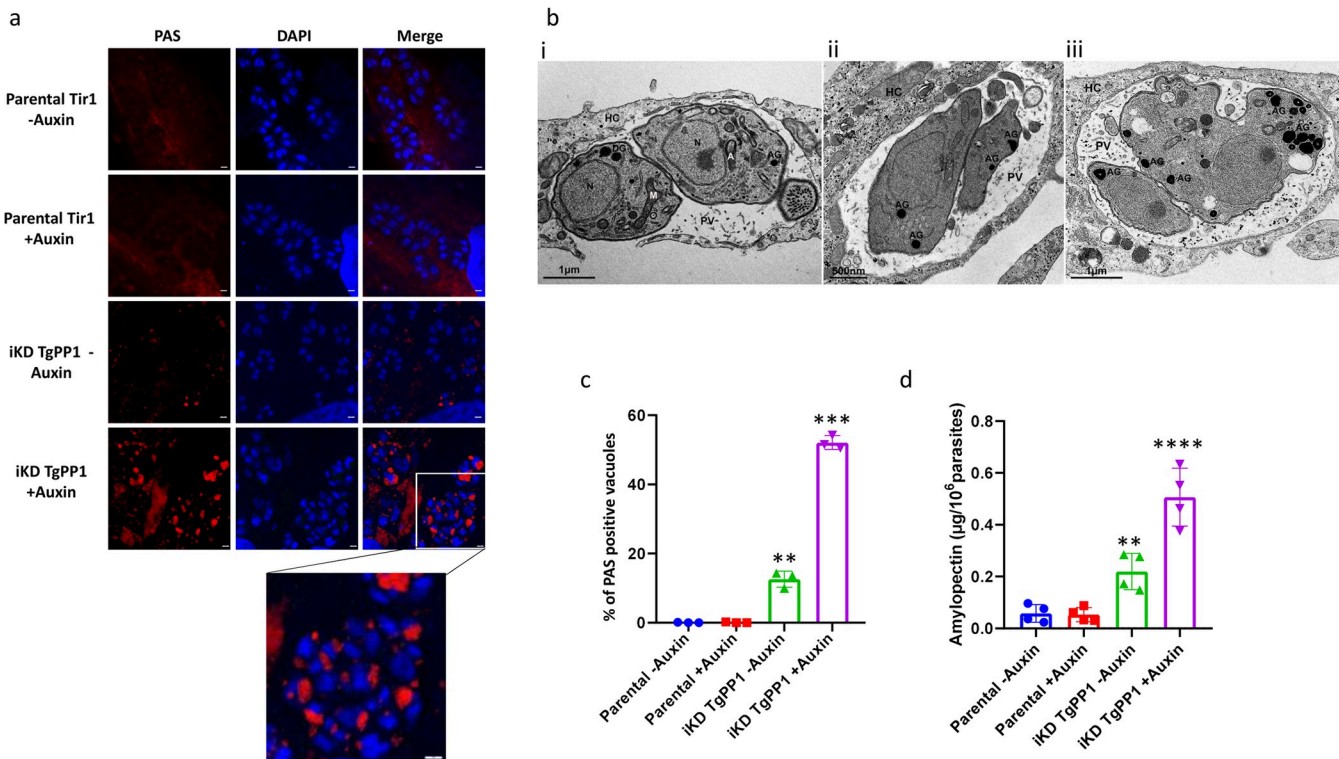

**Fig 4. Absence of TgPP1 results in the accumulation of amylopectin granules. (a)** Confocal imaging of the Parental Tir1 and iKD TgPP1 strain stained with PAS (red) in the absence and presence of auxin for 48 h. DAPI was used to stain the nucleus. Scale bar (1 μm) is located in the far-right corner of each image. **(b)** EM scan of the iKD TgPP1 parasites in the absence (i) or presence (ii, 24 h and iii, 48 h) of auxin treatment depicting the presence of AG in the iKD mutant. Starch was labeled using periodic acid. Note the accumulation of AG in the tachyzoites in (ii) and (iii). AG, amylopectin granules; DG, dense granules; N, nucleus; PV, parasitophorous vacuole; HC, host cell. Scale bar (1 μm or 0.5 μm) is demonstrated in the lower left region of each EM scan. **(c)** Bar graph representing the percentage of PAS positive vacuoles in the Parental Tir1 and iKD TgPP1 strains in the absence and presence of auxin for 48 h. A Student's *t* test was carried out. **$p < 0.01$, ***$p < 0.001$; mean ± SD ($n = 3$). **(d)** Quantification of the amount of amylopectin, as measured by biochemical assay, present within Parental Tir1 and iKD TgPP1 parasites in the absence and presence of auxin for 48 h. A Student's *t* test was carried out. ***$p < 0.001$, ****$p < 0.0001$; mean ± SD ($n = 3$). The data underlying this figure can be found in S1 Data. EM, electron microscopy; PAS, periodic acid–Schiff.

auxin for 48 h. There was a significantly increased percentage of PAS positive vacuoles (approximately 50%) in the iKD TgPP1 mutant in the presence of auxin when compared to the parental strain or the iKD TgPP1 mutant in absence of auxin (Fig 4D). However, as observed in Fig 4A, a significant number of vacuoles were PAS positive in the iKD TgPP1 mutant in absence of auxin (around 15%). We independently verified the accumulation of amylopectin in the iKD TgPP1 mutant in absence and presence of auxin by a biochemical dosage of amylopectin produced by this strain and the parental strain. In the absence and presence of auxin, the iKD TgPP1 mutant parasite possesses a significant amount of amylopectin (Fig 4E). The amount of amylopectin was significantly increased in presence of auxin in the iKD TgPP1 mutant (Fig 4E), indicating that the depletion of TgPP1 induces the accumulation of amylopectin in the parasite. Therefore, TgPP1 plays pleiotropic roles during the intracellular growth of the tachyzoite including regulation of the amylopectin steady-state levels.

## TgPP1 regulates the biology of the tachyzoite through dephosphorylation of a large set of proteins

Since TgPP1 was localized to the nucleus of the parasite, we investigated its potential role in gene regulation. For that we carried out RNA-sequencing analysis by examining changes in

the transcriptome of the iKD TgPP1 strain after treatment with or without auxin for 24 h from 4 biological replicates. Deseq2 was used to determine significant differential expression of genes based on an adjusted *p*-value cutoff of 0.05 and a minimum fold-change of 2 (S4A Fig). To our surprise, only a few genes were considered differentially regulated following the depletion of TgPP1 and none of them pass the minimum fold-change of 2 cutoff (S4A Fig and S1 Table). This suggested that TgPP1 had a minimal role in transcriptional regulation despite its nuclear localization.

We then investigated the role of TgPP1 in differential phosphorylation of proteins during the intracellular growth of the parasite. For that, the phosphoproteome of the iKD TgPP1 mutant parasite was investigated at 2 time points. A first time point, after 24 h of normal growth and then 2 h of auxin treatment, was aimed at discovering the direct targets of TgPP1, a short time period after its complete depletion (as measured by western blots). A second time point, after 24 h of auxin treatment was aimed at measuring the global effect of TgPP1 on the phosphoproteome. The presence of significantly hyper or hypo-phosphorylated peptides was identified in a minimum of 3 biological replicates.

After 2 h of auxin treatment, a total of 7,822 phosphorylation sites were identified (S2 Table), among the 4 biological replicates that were analyzed. Among these, 104 phosphopeptides exhibited significant differential phosphorylation with an FDR less than 0.01 (Fig 5A and S2 Table). Among the 104 phosphopeptides (representing 93 proteins), 37 proteins were considered significantly hyper-phosphorylated (40 phospho-peptides associated) and 56 proteins (64 phospho-peptides associated) were considered significantly hypo-phosphorylated (Fig 5A). We focused on the hyper-phosphorylated proteins that are likely the effect of the phosphatase depletion. We noticed that IMC1 was identified with the highest hyper-phosphorylation ratio (S2 Table). Moreover, the Apical Cap protein 2 (AC2, TGME49_250820), the CPH1-interacting protein 2 (CIP2, TGME49_257300) and TGME49_285850 are predicted to localize to the apical compartment or IMC [34] and were found to be hyperphosphorylated after 2 h of auxin treatment. Concordant with the TgPP1 nuclear localization, a high number of hyper-phosphorylated proteins (19/37) are predicted to localize to the nucleus. Among them, the DNA replication licensing factor MCM7 is likely involved in DNA replication. We also identified hyper-phosphorylation of 2 fitness-conferring kinases: CDPK6 [35] and TKL1 [36]. Moreover, we identified the hyper-phosphorylation of a trehalose phosphatase (TGGT1_297720), an uncharacterized protein containing a CBM20 domain, known to bind to starch. In the list of hypo-phosphorylated proteins, we also found 5 other IMC proteins such as IMC17, ISC1, PMCAA1, and uncharacterized IMC proteins (TGGT1_217510 and TGGT1_306190). Moreover, 4 putative kinases (CDPK2a, the cell cycle associated GSK (TGME49_265330), TGME49_225960, and TGME49_320000) and 1 phosphatase (TGME49_269460) were also found to be hypo-phosphorylated. A motif analysis of the differentially phosphorylated peptides indicates that serine is the main target of phosphorylation. There was no obvious enrichment of kinase substrates among these motifs (S5 Fig).

We then wanted to investigate further by studying the effect of TgPP1 depletion on the differential phosphorylation of phospho-peptides after a longer period of auxin treatment. After 24 h of auxin treatment, a total of 7,509 phospho-sites were identified. Out of these phospho-sites, 376 were considered as significantly differentially phosphorylated and corresponded to a total of 284 proteins (Fig 5B and S2 Table). The total number of hyper-phosphorylated peptides was 152, whereas the total number of hypo-phosphorylated peptides was 224, associated with 114 and 178 proteins, respectively. A striking number of IMC proteins (10/114) were discovered to be significantly hyper-phosphorylated such as IMC4, IMC18, IMC20, AC2, and ISC7. On the other hand, 17 proteins associated with the IMC or the apical compartment were identified as hypo-phosphorylated, suggesting a global perturbation of the phosphorylation

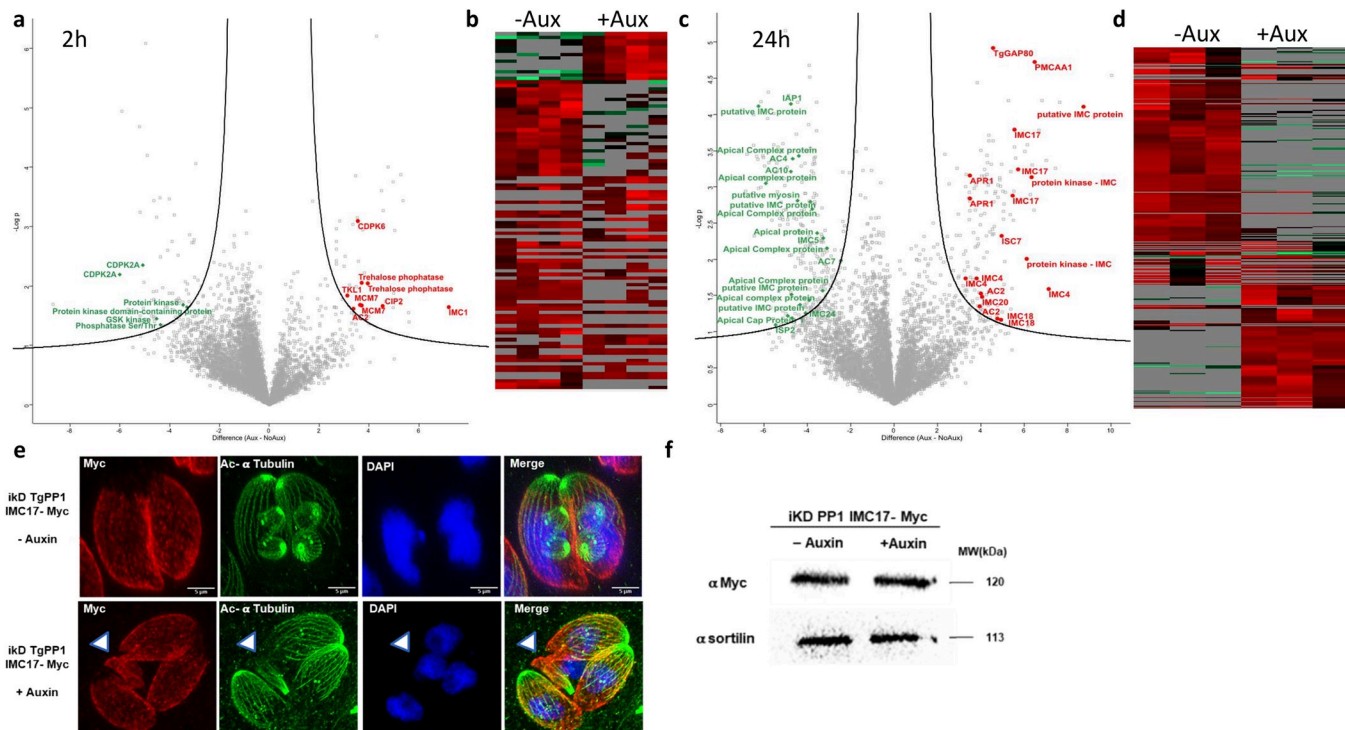

**Fig 5. TgPP1 depletion results in differentially phosphorylated proteins including a high number of IMC proteins. (a)** Volcano plot of the total phosphosites in the iKD TgPP1 mutant parasite after the treatment of auxin for 2 h resulting from phosphoproteomics analysis (*n* = 7,822). Selected proteins presenting hyperphosphorylated peptides in absence of TgPP1 are highlighted in red. Selected proteins presenting hypophosphorylated peptides are highlighted in green. See the full list in S2 Table. **(b)** Heat map of phosphosites which are differentially phosphorylated as a result of 2 h of auxin treatment in the iKD TgPP1 mutant. **(c)** Volcano plot of the total phosphosites in the iKD TgPP1 mutant parasite after the treatment of auxin for 24 h as demonstrated following phosphoproteomics analysis (*n* = 7,509). Selected proteins presenting hyperphosphorylated peptides in absence of TgPP1 are highlighted in red. Selected proteins presenting hypophosphorylated peptides are highlighted in green. See the full list in S2 Table. **(d)** Heat map displaying phosphosites which are differentially phosphorylated following the treatment of auxin for 24 h. **(e)** Expansion microscopy images of the iKD TgPP1 strain in absence and presence of auxin. The parasite IMC (IMC17, red) and cytoskeleton (acetylated tubulin, green) were labeled as well as the nucleus by DAPI (blue). A parasite with IMC formation defects is indicated by a white arrow. **(f)** Western blot image showing the unchanged level of expression of the TgIMC17 (IMC17-myc) protein in presence or absence of Auxin after 24 h. Sortilin was used as a loading control. IMC, inner membrane complex.

status of this compartment. As an illustration, a single IMC protein, IMC17, was found both hyper-phosphorylated at particular phospho-sites and hypo-phosphorylated at other phospho-sites. Of note, the Trehalose phosphatase (TGME49_297720) was also found hyper-phosphorylated after 24 h of treatment. Similar to what has been shown for the 2 h auxin treatment data set, there was no obvious enrichment of kinase substrates among the differentially phosphorylated peptide motifs (S6 Fig) although the GO analysis revealed an enrichment of proteins in kinase, enzyme regulator, and DNA-binding transcription factor activities (S6D Fig).

To assess the effect of the differential phosphorylation on IMC proteins, we tagged IMC17 in the iKD TgPP1 mutant and performed expansion microscopy after 24 h auxin treatment (Fig 6E). This experiment confirmed the IMC phenotypes described before.

When comparing our phosphoproteome analysis to that of Herneisen and colleagues [20], we identified a small overlap: around 22% of the protein present in our data set (21/93 at 2 h and 64/284 at 24 h) were common. Interestingly, among the hyperphosphorylated proteins that are common with our data sets, we found the apical/IMC proteins IMC4, AC2, AC13, and CIP2 and the centrosomal Cep250 protein (S2 Table). Among the hypo-phosphorylated proteins, the IMC/apical proteins AC4, IMC24, and AAP2 and the centromeric protein Cenp-C were present in both data sets.

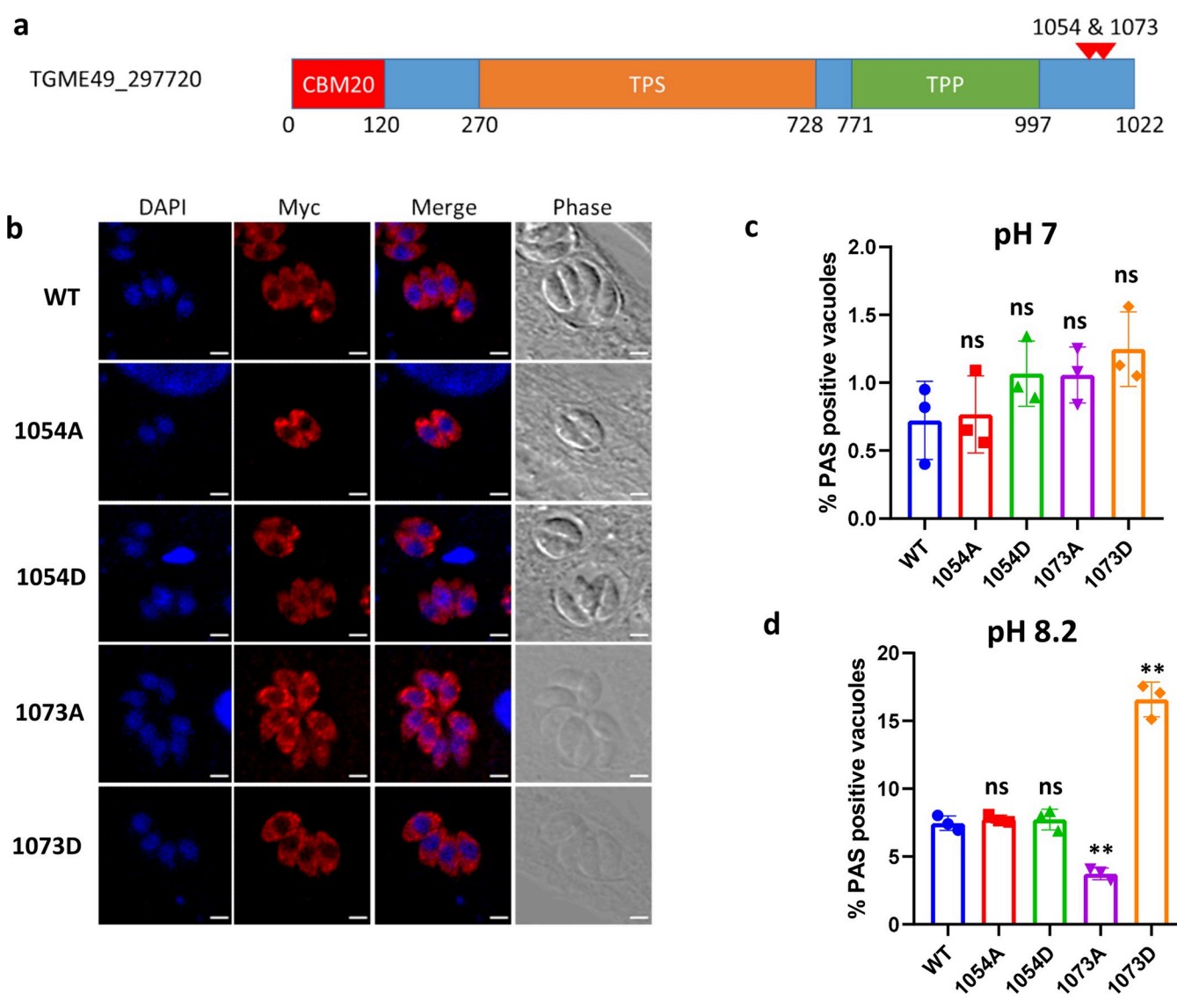

**Fig 6. The phosphorylation status of the TGGT1_297720 protein affects amylopectin steady-state levels.** (**a**) Schematic representation of the trehalose synthase-phosphatase protein (TGGT1_297720) that consisting of CBM20, trehalose synthase (TPS), and the trehalose phosphatase (TPP) domains. Differentially phosphorylated sites after TgPP1 depletion at Serine 1054 and Serine 1073 are indicated by red arrow heads. (**b**) Confocal imaging of the mutated TGGT1_297720 proteins. The myc tagged TGGT1_297720 protein is labeled with anti-myc antibodies (red). DAPI was used to stain the nucleus. Scale bar (2 μm) is indicated in the lower right corner of each individual image. The identity of the mutation is indicated on the left side. (**c**) Bar graph representing the percentage of PAS positive vacuoles at a pH of 7.0 in the WT and mutant 1054A, 1054D, 1073A, and 1073D parasites. A Student's *t* test was carried out. ns > 0.05; mean ± SD ($n$ = 3). (**d**) Bar graph representing the percentage of PAS positive vacuoles at a pH of 8.2 in the WT and mutant 1054A, 1054D, 1073A, and 1073D parasites. A Student's *t* test was carried out. ns > 0.05, **$p$ < 0.01; mean ± SD ($n$ = 3). The data underlying this figure can be found in S1 Data. PAS, periodic acid–Schiff; TPP, trehalose-6-phosphate phosphatase; TPS, trehalose-6-phosphate synthase.

As a control, we performed a global proteome analysis of the same samples that were treated for 24 h with or without auxin (S1 Table and S4B Fig) and did not identify significant differences in protein content in these samples, suggesting that TgPP1 mainly acts through modulating the phosphorylation status of proteins and does not affect global proteome content. These results were independently confirmed by a western blot of the iKD TgPP1 IMC17-myc strain showing that the IMC17 level of protein expression was unchanged in presence of auxin for 24 h (Fig 6F).

## TgPP1 dephosphorylation of the TGME49_297720 protein may influence starch metabolism

TgPP1 depletion influences starch metabolism (Fig 4). We identified the 2 differentially phosphorylated sites on the TGME49_297720 protein. This protein encodes for a starch binding domain (CBM20), a trehalose-6-phosphate synthase (TPS), and a trehalose-6-phosphate phosphatase (TPP) domain (Fig 6A). Both identified hyper-phosphorylated sites (Serine 1054 or Serine 1073) are situated outside of the putative enzymatic domains (Fig 6A). We explored whether these phospho-sites were significantly linked to the starch accumulation phenotype observed previously. For that, we produced 4 transgenic parasite mutants bearing single point mutations corresponding to Serine 1054 or Serine 1073 changed to an Alanine or Aspartate amino acid. A parental strain (RH ΔKu80) with the Trehalose phosphatase gene bearing a C-terminal Myc tag was used to generate the mutants using the CRISPR/Cas9 system and direct FACS sorting into 96-well plates (S7 Fig). To identify whether the WT or mutated fragment had been inserted into the correct locus of the genome, PCR fragments representing the targeted locus were amplified and sent for sequencing (S7 Fig). First, we verified that the mutation had no effect on the enzyme localization (Fig 6B). We then proceed to verify the effect of the induced mutation on accumulation of starch in these mutant parasites. For that, we quantified the number of PAS positive vacuoles in normal culture conditions (pH 7) or in conditions known to induce bradyzoite differentiation (pH 8.2), the bradyzoite form accumulates amylopectin. In normal culture conditions, there was no significant difference between the WT and the mutant parasites, when measuring the percentage of PAS positive vacuoles. However, in bradyzoite inducing conditions (pH 8.2), the percentage of PAS positive vacuoles significantly decreased in the 1073A mutant, whereas it significantly increased in the 1073D mutant compared to their wild-type counterpart. These results suggest that the phosphorylation status of the Trehalose phosphatase is linked to the ability of the parasite to accumulate starch, mimicking the phenotype observed after TgPP1 depletion.

## Discussion

Phosphorylation is a widespread posttranslational modification in apicomplexan parasites [6]. In *T. gondii*, it is involved in the regulation of the tachyzoite cell cycle [37] and of starch metabolism [18]. Here, we explored the pleiotropic phenotypes caused by the depletion of the TgPP1 protein. A previous study investigated TgPP1 roles by focusing on the late stage of the tachyzoite cycle and examining the critical roles of this protein in egress and motility of the parasite [20], a feature that appeared common with PfPP1 [21]. In our study, we focused on the division of the tachyzoite and showed that TgPP1 is involved in a wide range of molecular mechanisms including nuclear and organelle segregation. Therefore, we can conclude that TgPP1 is a crucial phosphatase that regulates both the intracellular cell cycle (this study) and the extracellular mobility [20]. Genetic manipulation of this locus has proven challenging and the resulting independent mutants [20] have shown a decreased fitness in conditions where TgPP1 was still expressed, suggesting that the activity of this protein is highly important for the parasite. The mutant produced in this study, although noticeably affected in absence of auxin, showed stronger phenotypes under auxin treatment.

Among the phenotypes observed, IMC formation was one of them and, as confirmed by IFA and EM. Differential phosphorylation in absence of TgPP1 of multiple IMC targeted proteins such as IMC1, IMC4, IMC5, IMC17, IMC18, IMC20, IMC24, ISP1, and ISC1 is a hallmark of the phospho-proteomics experiments we performed. This was only partially confirmed by the previous study where IMC4, AC2, AC13, and CIP2 were found differentially phosphorylated, although technical and analytical differences impede a direct comparison

between the 2 data sets. A short time after TgPP1 depletion, TgIMC1 was found to be hyper-phosphorylated at the threonine 62, with the greatest amplitude when comparing the auxin-treated samples to the non-treated ones. IMC proteins are targeted sequentially to the IMC and form a rigid meshwork onto the flattened alveolar sacs [38]. Posttranslational modifications can influence the function and assembly of the IMC proteins. For example, TgIMC1 undergoes proteolytic cleavage and the cleaved form is associated with filament rigidity [39]. Numerous phosphorylation sites have been detected on IMC proteins although their biological significance remained to be investigated. Of note, assembly and rigidity of intermediate filament proteins (e.g., nuclear lamina network) in other organisms have been shown to be regulated through phosphorylation [40]. Phosphorylation on serine and threonine residues would generally promote disassembly and dephosphorylation increasing stability of the intermediate filament [41]. In a genome-wide screen for kinase function, among the 15 kinases that were essential for tachyzoite growth, at least 9 showed phenotypes linked to IMC formation [9], indicating the importance of phosphorylation in this process. After TgPP1 depletion, we have found a global collapse of the IMC and hyper-phosphorylation of multiple IMC proteins that suggest that the IMC network assembly and stability may be controlled by similar phosphorylation mechanisms as for the assembly of other intermediate filaments. Of note, the phosphorylation status of the IMC proteins does not seem to control their overall stability. Indeed, the global proteome was unaffected after 24 h auxin treatment although IMC defects were seen in a majority of vacuoles. In support of this hypothesis, the amount the TgIMC17 protein remained unchanged after depletion of TgPP1, although its phosphorylation status is drastically changed (hyperphosphorylated and hypophosphorylated) at several amino acid locations.

The specific role in the assembly of the IMC of the TgIMC1 Thr62 phosphorylation would be of interest to investigate in the future, although individual phosphorylation sites on TgIMC12 were not shown to impact the IMC assembly [42]. Rather than the limited phosphorylation status of individual IMC proteins, TgPP1 depletion seems to induce a global imbalance in the phosphorylation status of IMC proteins that may be the cause of the visual phenotype observed. This is reminiscent to the PfPP1 mutant phenotype for which the IMC failed to form when depletion of the protein was achieved mid-intraerythrocytic developmental cycle [21].

Phosphorylation of TgGAP45 has been shown to control its assembly to the MyoA-glidosome [43], a structure essential for the extracellular movement of the parasite. Interestingly, TgGAP80, a protein belonging to the TgGAP45 protein family, was found to be hyperphosphorylated after TgPP1 depletion. Moreover, IAP1, a protein associated with TgGAP80 is also differentially phosphorylated in presence of auxin. Both proteins are part of the MyoC-glideosome, a complex associated with IMC proteins that localizes at the basal pole of the parasite [44]. However, the mutation of individual phosphorylation sites on the TgGAP45 protein did not affect glideosome assembly and parasite growth, indicating the modest contribution of individual phosphorylation site to the glideosome or pellicle assembly [45].

Taken together, these data indicate that the phosphorylation status of IMC proteins might be an important determinant of their ability to assemble and rigidify of the IMC network. The structural morphology of the iKD TgPP1 mutant is impacted as was indicated through EM. In the presence of auxin, the usual bow-shaped tachyzoite is absent and instead an abnormally shaped tachyzoite takes its place. Despite that the plasma membrane remains intact, it appears to exhibit unusual curvatures and invaginations, this might possibly be due to the collapsed IMC structure which is absent underneath it. In normal conditions, the tachyzoite structure is maintained through IMC proteins resulting in tachyzoite pellicle strength. Our data indicates that TgPP1 influence on IMC may be independent from its potential role in modifying centrosome activity.

The presence of nuclear and organelle segregation phenotypes after depletion of TgPP1 suggests that this phosphatase may have targets that are organizing cell division in the parasite. Although the centrosome division remains mainly unaffected in absence of TgPP1, TgCep250 was identified as differentially phosphorylated in our data set and in Hernesien and colleagues [20]. TgCep250 mutants led to a high number of parasites presenting nuclear segregation defects [46], a phenotype that is also observed in the iKD TgPP1 parasite in presence of auxin. Overall, the mitosis defects after depletion of TgPP1 points toward a desynchronization of the budding and nuclear cycle leading to production of parasites without nucleus or nucleus without parasite body. Detailed examination of mitosis after depletion of TgPP1 indicates that metaphase may be slowed in this mutant but mitosis eventually processed. This in contrast of the function observed in *Plasmodium* PP1 for which DNA replication was strongly affected [21,23].

RNA-seq revealed that TgPP1 depletion did not change the transcriptome of the parasite after 24 h of auxin treatment. Despite TgPP1 nuclear localization and the presence of differential phosphorylation of ApiAP2 transcription factors in the phospho-proteome at 24 h auxin treatment, the phenotypes observed are not the consequence of differential transcript expression or proteome expression. ApiAP2 posttranslational modifications have been suggested to modify their activity [2] but the data presented here does not support this hypothesis.

The formation of amylopectin granules in the tachyzoite following the depletion of TgPP1 was a striking finding. This suggests that TgPP1 has a role in the regulation of amylopectin steady-state levels during tachyzoite proliferation. Phosphorylation has been shown to regulate amylopectin metabolism in *T. gondii*. TgCDPK2 phosphorylates a wide range of enzymes that are involved in this process [18] among which a glycogen phosphorylase, an alpha-glucan water dikinase, and a pyruvate phosphate dikinase that are known to bind to amylopectin through CBM20 domains [47]. Mutating the phosphorylation sites of glycogen phosphorylase was later shown to phenocopy the TgCDPK2 knock-out strain [48] suggesting a crucial role for phosphorylation in regulating the steady-state levels of amylopectin in this parasite. A phosphatase was also recently implicated in this pathway: TgPP2A contributes to the regulation of amylopectin metabolism via dephosphorylation of TgCDPK2 at a particular site (S679) [16] suggesting a pivotal role for TgCDPK2 in amylopectin accumulation. We found that a protein annotated as a Trehalose phosphatase was differentially phosphorylated in absence of TgPP1. Mutagenesis of these phosphosites recapitulates partially the TgPP1 depletion phenotype with amylopectin accumulation being detected after induction of bradyzoite differentiation. This may be due to the fact that single phosphosite mutants could not recapitulate the extend of the perturbation that are caused to the parasite environment when TgPP1 is depleted. Moreover, our phosphoproteomic data may not be exhaustive as other phosphosites on this protein have been discovered [6] and could play a role in the regulation of the activity of this protein. Interestingly, the TGME49_297720 protein was not listed as the potential target of TgCDPK2 or TgPP2A suggesting that it acts through an independent pathway that also controls the metabolism of amylopectin. In favor of this hypothesis, accumulation of amylopectin was only observed in the cytoplasm of the parasites lacking TgPP1 while it was observed both in the cytoplasm and residual body of parasites mutated for TgCDPK2 [47] or TgPP2A [17], suggesting a different regulation mechanism. TGME49_297720 encompasses a starch-binding domain (CBM20) that further supports the hypothesis of a function linked to starch metabolism regulation. This protein is annotated as a Trehalose phosphatase. However, its sequence encodes a TPS and a TPP domain. Dual activity enzymes are found in some bacteria and fungi [49]. The TGME49_297720 protein may therefore be misannotated as a trehalose phosphatase and should be annotated as a bifunctional TPS-TPP protein. The presence of this dual-activity enzyme indicates that *T. gondii* may have the ability to produce trehalose from UDP-Glucose

and Glucose-6-Phosphate as it is the case for numerous eukaryotes. However, these activities have to be confirmed experimentally. Trehalose is a nonreducing disaccharide consisting of 2 glucose subunits with an α,α′-1,1′-glycosidic bond. This carbohydrate occurs in a wide range of species and is synthesized by bacteria, fungi, plants, and various invertebrates. Trehalose has roles in development, stress tolerance, energy storage, and the regulation of carbon metabolism in plants and fungi [50]. In stress conditions, trehalose is also known to act as a stabilizing factor for protein structures [51]. Starch accumulation is linked to stress-induced differentiation in *T. gondii*; therefore, trehalose may be acting as a chaperone during this process. Alternatively, trehalose-6-phosphate, the product of the enzymatic reaction catalyzed by TPS, is known as a signal molecule in plants [50]. Trehalose-6-phosphate acts by inhibiting starch degradation by preventing the early steps in the starch degradation pathway within the chloroplast, although this effect may be indirect through the regulation of kinases and phosphatases that control the activity of the enzymes responsible for starch degradation [52]. Further investigations are warranted to discover if trehalose-6-phosphate plays a similar role in regulating starch degradation in the parasite. While trehalose or trehalose-6-phosphate have not been identified so far in the parasite [53], the data presented here suggests that it may have a role in regulating starch metabolism in the parasite.

Overall, we showed that TgPP1 acts through the phosphorylation of a wide range of targets to regulate a large set of molecular mechanisms that are essential for the completion of the tachyzoite cell cycle. Our study puts an emphasis on the role of phosphorylation in the IMC protein network assembly and stability. It also confirms the importance of phosphorylation in regulating the amylopectin steady-state levels in the parasite and suggests the involvement of a new regulatory pathway involving a new enzyme in this pathway.

## Supporting information

**S1 Table. RNA-seq analysis of the iKD TgPP1 strain in absence and presence of auxin for 24 h.**
(XLSX)

**S2 Table. Phosphoproteomics results for the iKD TgPP1 strain in absence and presence of auxin for 2 h or 24 h.** Quantification of each differentially phosphorylated peptide is listed along with the targeted amino acid and the kinase family potentially phosphorylating the peptide.
(XLSX)

**S3 Table. Results of the global proteome analysis of the iKD TgPP1 strain in absence and presence of auxin for 24 h.**
(XLSX)

**S1 Data. Raw data for figures: Figs 1D, 2C, 2D, 3C, 4C, 4D, 6C, 6D, S1B, S1C, S2D, S2E, S3B, S3D, S4A and S4B.**
(XLSX)

**S1 Raw Images. Uncropped version of the western blots and gel images.**
(PDF)

**S1 Fig. iKD TgPP1 mutant construction. (a)** PCR verifying integration of the HXGPRT-2TA-AID-Ty cassette at the correct genome locus of the iKD TgPP1 mutant. A band corresponding to 1536 using iKD TgPP1 genomic DNA confirms cassette integration compared to using WT genomic DNA. **(b)** Growth assay of the Parental Tir1 and iKD TgPP1 mutant strains in the absence and presence of auxin treatment for 24 h. A Student's *t* test was

performed, ns > 0.05, \*\*$p < 0.01$; mean ± SD ($n = 5$). **(c)** Bar graph indicating plaque size produced by the Parental Tir1 and iKD TgPP1 strain in the presence and absence of auxin. Plaque size was determined by measuring the percentage of lysed surface of the plaque assay. Three independent experiments were carried out. A Student's *t* test was performed, \*\*$p < 0.01$; mean ± SD ($n = 3$). The data underlying this figure can be found in S1 Data. (PDF)

**S2 Fig. iKD TgPP1 demonstrates a collapsed IMC verified through TgGAP45 and TgISP1 labeling. (a)** Confocal imaging of the Parental Tir1 and iKD TgPP1 strains labeled with anti-TgGAP45 (red) in the presence and absence of auxin treatment. DAPI was used to stain the nucleus. Scale bar (1 μm) is indicated in the lower right corner of each individual image. **(b)** Confocal imaging of the Parental Tir1 and iKD TgPP1 strains labeled with anti-TgEno2 (red) and anti-TgISP1 (green) in the presence and absence of auxin treatment. DAPI was used to stain the nucleus. Scale bar (1 μm) is indicated in the lower right corner of each individual image. **(c)** Confocal imaging of the Parental Tir1 and iKD TgPP1 strains labeled with TgEno2 (red) and TgIMC1 (green) in the presence and absence of auxin treatment. DAPI was used to stain the nucleus. Scale bar (1 μm) is indicated in the lower right corner of each individual image. **(d)** Bar graph representing the percentage of Parental Tir1 and iKD TgPP1 vacuoles possessing a collapsed IMC by using anti-TgIMC1 antibodies for labeling the IMC in the absence and presence of auxin treatment for 48 h. A Student's *t* test was carried out, \*\*\*$p < 0.001$; mean ± SD ($n = 3$). **(e)** Bar graph representing the number of parasite observed by EM with inner membrane defects in the iKD TgPP1 strain in absence of auxin ($n = 53$), after 24 h of auxin treatment ($n = 44$) or after 48 h of auxin treatment ($n = 46$). The data underlying this figure can be found in S1 Data. (PDF)

**S3 Fig. Conditional depletion of TgPP1 has a qualitative impact on the outer core centrosome. (a)** Confocal imaging of Parental Tir1 and iKD TgPP1 parasites in the absence and presence of auxin treatment for 48 h labeled with anti-TgCentrin1 antibodies (green). DAPI was used to stain the nucleus. Scale bar (1 μm) is indicated in the lower right corner of each image. **(b)** Bar graph demonstrating TgCentrin1: nucleus ratio of Parental Tir1 and iKD TgPP1 in the absence and presence of 48-h auxin treatment. A Student's *t* test was performed, ns: $p > 0.05$; mean ± SD ($n = 3$). For each nucleus, the number of centrin dots is accounted independently from their size. Overall, more than 100 individual nuclei are counted for each biological replicate. **(c)** Representative expansion microscopy images of the iKD TgPP1 parasites in presence of auxin treatment for 24 h labeled with anti-Nuf2 (kinetochores, red) and anti-acetyl Tubulin (cytoskeleton, green) antibodies. DAPI was used to stain the nucleus. Scale bar (5 μm) is indicated in the lower right corner of each image. **(d)** Bar graph comparing the number of parasite undergoing Metaphase of Anaphase of the iKD TgPP1 in the absence ($n = 414$) and presence ($n = 426$) of 24-h auxin treatment. A $Chi^2$-test was performed, \*\*\*\*: $p < 0.001$; the number of parasites scored is indicated on top of the graph. The data underlying this figure can be found in S1 Data. (PDF)

**S4 Fig. RNA-sequencing and proteome analysis at 24 h of auxin treatment displays only a few differentially regulated genes. (a)** Volcano plot demonstrating the differentially expressed genes analyzed from RNA-sequencing of the iKD TgPP1 mutant parasite treated with auxin for 24 h. Differential expression is based on the analysis of 3 biological replicates. Statistically significant differentially expressed genes are indicated in red. However, these do not pass the +/− 1 $\log_2$ expression ratio criteria. **(b)** MA (Bland–Altman) plot demonstrating

the differential proteome content in iKD TgPP1 mutant parasites after 24 h of auxin treatment compared to iKD parasite grown in the absence of auxin (control). Statistically significant differentially expressed proteins are indicated in red. The data underlying this figure can be found in S1 Data.
(PDF)

**S5 Fig. Analysis of the phosphoproteome after 2-h treatment.** (A) Proportional representation of phosphorylation sites on serine, threonine, and tyrosine for total phosphorylation sites (TOTAL), significant down-regulated phosphorylation sites (DOWN) and significant up-regulated phosphorylation sites (UP). (B) Bioinformatics analysis of potential kinases and/or binding partners based on the surrounding sequence of altered phosphorylation sites revealed the 10 most common kinase and/or binding partner substrate motifs detected among significant phosphoproteins. (C) Sequence logos for significant phosphorylation motifs (left for down-regulated phosphorylation sites, right for up-regulated phosphorylation sites) where the phosphorylated residue (S or T) is centered.
(PDF)

**S6 Fig. Analysis of the phosphoproteome after 24-h auxin treatment. (A)** Proportional representation of phosphorylation sites on serine, threonine, and tyrosine for total phosphorylation sites (TOTAL), significant down-regulated phosphorylation sites (DOWN) and significant up-regulated phosphorylation sites (UP). (**B**) Bioinformatics analysis of potential kinases and/or binding partners based on the surrounding sequence of altered phosphorylation sites revealed the 10 most common kinase and/or binding partner substrate motifs detected among significant phosphoproteins. (**C**) Sequence logos for significant phosphorylation motifs (left for down-regulated phosphorylation sites, right for up-regulated phosphorylation sites) where the phosphorylated residue (S or T) is centered. (**D**) A table representing GO enrichment analysis on the hyperphosphorylated proteins. The analysis was performed using the Toxodb tool.
(PDF)

**S7 Fig. Schematic representation of the construction of point mutant of the TGGT1_297720 gene.**
(PDF)

## Author Contributions

**Conceptualization:** Mathieu Gissot.

**Formal analysis:** Asma Sarah Khelifa, Marcia Attias, Ida Chiara Guerrera, Wanderley De Souza.

**Funding acquisition:** Mathieu Gissot.

**Investigation:** Asma Sarah Khelifa, Maanasa Bhaskaran, Tom Boissavy, Thomas Mouveaux, Tatiana Araujo Silva, Cerina Chhuon, David Dauvillee, Emmanuel Roger.

**Methodology:** David Dauvillee.

**Resources:** Mathieu Gissot.

**Supervision:** Mathieu Gissot.

**Validation:** Asma Sarah Khelifa, Maanasa Bhaskaran.

**Visualization:** Maanasa Bhaskaran.

**Writing – original draft:** Asma Sarah Khelifa, Mathieu Gissot.

**Writing – review & editing:** Maanasa Bhaskaran, Mathieu Gissot.

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
