## [Editor Report · Decision Letter 0]

22 Aug 2023

Dear Dr. Gissot, 

Thank you for submitting your manuscript entitled "The PP1 phosphatase exhibit pleiotropic roles controlling both tachyzoite cell cycle and amylopectin-steady state levels in Toxoplasma gondii." for consideration as a Research Article by PLOS Biology.

Your manuscript has now been evaluated by the PLOS Biology editorial staff and I am writing to let you know that we would like to send your submission out for external peer review.

Once your full submission is complete, your paper will undergo a series of checks in preparation for peer review. After your manuscript has passed the checks it will be sent out for review. To provide the metadata for your submission, please Login to Editorial Manager (https://www.editorialmanager.com/pbiology) within two working days, i.e. by Aug 24 2023 11:59PM.

Kind regards,

Paula

---

Senior Editor

PLOS Biology

---

## [Decision Letter · Decision Letter 1]

20 Oct 2023

Dear Dr. Gissot,

Thank you for your patience while your manuscript "The PP1 phosphatase exhibit pleiotropic roles controlling both the tachyzoite cell cycle and amylopectin-steady state levels in Toxoplasma gondii." was peer-reviewed at PLOS Biology. Your manuscript has been evaluated by the PLOS Biology editors, an Academic Editor with relevant expertise, and by several independent reviewers.

As you will see in the reviewer reports, which can be found at the end of this email, although the reviewers find the work potentially interesting, they have also raised a substantial number of important concerns. Based on their specific comments and following discussion with the Academic Editor, it is clear that a substantial amount of work would be required to meet the criteria for publication in PLOS Biology. However, given our and the reviewer interest in your study, we would be open to inviting a comprehensive revision of the study that thoroughly addresses the reviewers' comments. Given the extent of revision that would be needed, we cannot make a decision about publication until we have seen the revised manuscript and your response to the reviewers' comments. Your revised manuscript would need to be seen by the reviewers again, but please note that we would not engage them unless their main concerns have been addressed. 

Having discussed the reviews with the Academic Editor, we think you should do some analysis to strengthen the associations iinferred between hyper phosphorylated proteins and the pp1 KD. The large number of hypophosphorylated proteins does indicate indirect effects of pp1 kd on phosphorylation so it is not possible to infer a direct role of pp1 in dephosphorylation of the hyper phosphorylated proteins in the kd. The specific experiments asked by reviewer #2 to be repeated could relate specific proteins with aberrant phosphorylation to the IMC and organelle segregation defects. Reviewer #3 makes the point that the cell cycle has not really been assessed, so we think that you could try and directly assess the effects of pp1 kd on cell cycle, Reviewer #1 says you could do so by additional imaging. You can also find the comments from the academic editor at the end of this letter.

We appreciate that these requests represent a great deal of extra work, and we are willing to relax our standard revision time to allow you 6 months to revise your study. Please email us (plosbiology@plos.org) if you have any questions or concerns, or envision needing a (short) extension.

**IMPORTANT - SUBMITTING YOUR REVISION**

*Resubmission Checklist*

*Published Peer Review*

*PLOS Data Policy*

*Blot and Gel Data Policy*

Sincerely,

Paula

---

Senior Editor

PLOS Biology

REVIEWS:

Reviewer #1: Apicomplexan cell biology.

Reviewer #2: Apicomplexan cell signalling.

Reviewer #3: Toxoplasma.

Reviewer #1: In this manuscript, the authors investigate the role of PP1 phosphatase during the lytic cycle of Toxoplasma gondii. Using the Auxin Inducible Degron, the enzyme was previously shown to regulate Ca2+ uptake to promote parasite motility (PMID35976251). Here using a similar strategy, the authors identify different requirements for PP1 during the lytic cycle including for the formation of the inner membrane complex, cell division and amylopectin levels. They further investigate the role of PP1 by determining the transcriptome, proteome and phosphoproteomes at two time points following invasion. While little differences are observed at the transcriptome and proteome levels, significant changes are observed in the mutant phosphoproteome. This includes higher abundance of 40 phosphopeptides, some of them mapping on IMC proteins, and 2 phosphopeptides mapping on a putative multidomain trehalose phosphatase. Phosphomimetic or phosphonull mutants did not show a phenotype in normal culture conditions but showed a correlation between putative phosphorylation of this protein with the presence of Periodic Acid Shiff (PAS) positive vacuoles at pH8.2, a growth condition that induces bradyzoite differentiation.

Major comments

Introduction - The generalization from Toxoplasma to Apicomplexa leads to inaccurate statements. I would strongly suggest the authors to generally use Toxoplasma instead of Apicomplexa unless for specific comparisons.

There is no formal analysis or comparison with the phenotypes observed here and the phenotypes previously described in PMID35976251. These phenotypes do not overlap and look at time discrepant. For example, the IMC defects were either missed or absent in the previous study that only referred to delayed intracellular growth. While I do not want to question the validity of the results of one or the other study, it would be important to clearly address this point to infer why the described phenotypes are different and somehow discrepant.

It would be important to better describe the proteome results and to get an idea about their depth. This also holds true for the phosphoproteome with no GO term enrichment nor motif analysis. This would help to get a better idea of the global changes. Of note many proteins highlighted are not IMC proteins and the authors should be careful and more comprehensive in their description of the dataset. 

Could the authors discuss the possible reasons explaining the higher abundance of phosphopeptides mapping on IMC protein? Would this be due to mislocalization of these proteins in the absence of IMC? Or is the defective IMC formation due to trafficking defect for example? The results presented here are correlative but give limited insights in explaining the cause of the observed defects.

Similarly, the link between PP1 and the putative trehalose phosphatase is currently difficult to establish in absence of direct comparisons with PP1 under similar experimental conditions (pH of the culture in this case). In addition, multiple mutants have been previously associated with altered amylopectin metabolism but with different phenotypes including the localization of storage granules. This is here not shown for the trehalose phosphatase mutants. Of note, it is a bit worrying to see different amino acid numbers for the mutated sites across the panels and figure legends. Finally, while I am not an expert in metabolism, could the authors speculate in details about the link between a trehalose phosphatase and amylopectin accumulation?

Minor comments

line 40-41 - I would use "has been used" instead of "emerged" - in the case of amylopectin, Toxoplasma does not seem to be a model for Plasmodium

line 59 and after - Phosphorylation is generally regarded as a post-traductional modification not post-transcriptionnal modification even if this is technically correct

line 64 - given the importance of the centrosome in the result section, the specificities of the Toxoplasma centrosome could deserve a bit more attention in the introduction

line 381 - I would replace "essential" by "important"

Fig S3 - please define better how the centrin to nucleus ratio is determined. As a non-expert, I see less centrin dots in the lower panel unless the authors consider the tiny dots as normal centrin staining. In addition, the determination of the number of nuclei numbers seems non-trivial given the "diffuse" DAPI signal - could the authors better describe the criteria used for this quantification?

line 428 - the transition to amylopectin seems to come out of nowhere (as far as I know, investigating amylopectin levels is not a systematic phenotyping in the field of apicomplexan kinases/phosphatases). Are amylopectin granules already visible in the EM images of Figure 2, this could justify more attention?

line 438 - how do the authors know that the observed structures seen by EM are amylopectin granules?

line 481 and elsewhere - I am not sure that the terminology hyper/hypophosphorylated is correct. Does it mean that more or less sites are phosphorylated on the detected peptides? As far as I understand the authors measure the relative abundance of phosphorylated peptides, if I am wrong please define better what hypo/hyperphosphorylated means.

line 569 - please also discuss PMID24779470 that showed that deletion of twelve phosphorylation sites within GAP45 did not impact its dual function in motor recruitment and pellicle integrity. This may give a more balanced view of our exact knowledge on how phosphorylation regulates biological processes in Toxoplasma.

Line 599 - the phenotype described for PP1 seems different from the CDPK2 phenotype, it would be important to discuss these differences

Reviewer #2: 

The authors have presented their interesting finding on conditional knockdown of PP1 in Toxoplasma, an apicomplexan parasite. The role of PP1 in Toxoplasma has also been investigated in number of studies earlier that authors do mention in their introduction.

Though it is mainly concentrated on Toxoplasma but they completely ignore the various recent studies in closely related apicomplexan Plasmodium falciparum both in their introduction and the discussion with respect of number of function they investigated here (PMID: 32669539, PMID: 34145386; PMID: 35336160; PMID: 33036936; MID: 37628837). It is surprising as the group is working in the same institution where PP1 has been studied widely in Plasmodium by Khalife group. So to complete ignore work in closely related apicomplexan Plasmodium is quite intriguing. 

They do not review and mention about the similar finding in Plasmodium in terms of IMC and other defects of PP1 knockdown. So the novelty and significance of the finding is bit subdued because of these recent findings. Fully appreciate that it has been just been repeated in Toxoplasma. Many of the technical approaches are also similar to what has been done in Plasmodium. The only additional information in terms of function of PP1 in Apicomplexa that is important is the its role in starch production that has not been shown earlier. 

It is not clear from thestudy presented as to what is the focus of the authors to whether they want to analyse the function of PP1 during cell cycle, cell division or the starch metabolism. The different part of the study does not really focus on any one aspect in more detail. It is clear from various studies on PP1 in eukaryotes that PP1 has varied roles but it would have been better if the authors had focussed on any one aspect in more detail here. 

The few suggestions to have some clear focus either cell cycle, cell division or starch metabolism with PP1 in Toxoplasma are as:

1. It will be informative if the authors present some good high quality images with either expansion microscopy or the zoom of the confocal images. Please shows the clear defect in the cell division and segregation with some markers like for kinetochore Ndc80, gamma tubulin etc for defect in division and segregation. This has been performed very elegantly in Toxoplasma in many recent studies. This can performed with different phases of cell cycle to get this part bit more informative. It will be interesting to know if the chromosome segregation or the DNA replication both affected. From the images presented in the manuscript it is not very clear about the defect in organelle division. 

2. Some clear markers for IMC and glideosome component could be studied for as they show that in their phosphoproteome data that these components are affected. Can they look for the component of the IMC complex or glideosome complex that is affected in PP1 knockdown parasite and at what stage during intracellular stage.

3. For the starch metabolism part- can they please perform lipodomics to show what component of these starch granule are affected. 

Some points on the figures etc:

Fig1 d. It seems at 48h there is still growth. What happens after 48 hours?

Fig2 How many cells counted. Does not look like DNA has even replicated. Would be great to quantify DNA here before saying missegregation phenotype. Also for EM images how many were counted with he defects and did all showed the same defect. It will be good to look in depth for defect with in the nuclear compartment as well with some markers.

Fig3. phase images are not clear and may be some zoom of the regions are given or expansion microscopy is performed with these marklers to show clear defect in Fig3a. 

Fig4 Again the phase are not clear to understand the cell structure. The quantification EM images will be good. At what stage of cell cycle these images were captured

Fig5. Labelling are too small to see the protein names and axes.

Reviewer #3: This manuscript from Khelifa and colleagues describes multiple cellular functions of an essential phosphatase in the eukaryotic pathogen Toxoplasma gondii. They have engineered an inducible knockdown parasite line to evaluate the role of TgPP1 during tachyzoite replication. Using a phosphoproteomics approach, they have identified a number of PP1 targets involved in many cellular processes, including regulation of amylopectin levels. As accumulation of amylopectin is a hallmark of Toxoplasma chronic stage infection and little is known about how the parasite regulates this metabolic process between acute and chronic stages, identification of regulators like PP1 and its target TGME49_297720 will be of interest to the field. The experiments have been performed and analyzed appropriately, the data has been well presented (apart from a few small errors noted below) and the conclusions made by the authors are supported by the reported results.

This is not the first report of an inducible PP1 knockdown in Toxoplasma tachyzoites, so although the impact is somewhat diminished, this study does yield novel findings on the role of this phosphatase in organelle division and segregation, parasite structure and amylopectin metabolism.

Major comments

I recommend that the authors consider removing "control of tachyzoite cell cycle" from the title of the paper. There is no evidence presented in the manuscript that PP1 directly regulates cell cycle progression. Perhaps something like "The PP1 phosphatase exhibits pleiotropic roles controlling both daughter cell formation and amylopectin-steady state levels in Toxoplasma gondii" is a little more representative of the data in the paper?

Figure 1d. An average of four parasites per vacuole after 48hr of growth for both parental and iKD TgPP1 parasites, with or without auxin. This is very low. Does this represent parasites that have egressed from the first round of intracellular replication and reinvaded? Or have the parasites been pretreated with auxin for a period, and then inoculated into plates with coverslips for further culturing up to 48hr of total treatment time? A more detailed description of the experimental set up of this growth assay is needed to clarify this.

Another recent paper (ref 14, Herneisen et al) reported on phenotypes of Toxoplasma tachyzoites during PP1 inducible knockdown and included a phosphoproteomics approach like the one reported here. Although this paper has been referenced a couple of times in this manuscript, a more thorough discussion of the important differences between the two studies should be included. Did the phosphorylation marks identified in both studies display the same trend i.e. increase or decrease during PP1 knockdown? There are some important technical differences in the experimental approaches used in these studies (differences in timing, perturbing calcium levels etc.) so it would be relevant and helpful to interested readers to highlight what is different with this study and what has been learned from this approach over the previous one. 

Minor comments

Line 23 eukaryotic, not eucaryotic

Line 47 consumption not consummation

Line 72 it would be more precise to say that homologues of TgPP1 have been studied in other organisms, rather than TgPP1 is a phosphatase that has been studied in other organisms.

Line 81-83 The reference for interaction of PP1 with LRR1 (Daher et al 2006) refers to a paper on Plasmodium falciparum. There is no experimental evidence for interaction between these two protein homologues in Toxoplasma. 

Line 482 should say "hyperphosphorylated" not "hypophosphorylated"?

Line 523 refers to supplementary figure 5, but a supplementary figure 5 has not been provided.

Figure 6b. Labels on the IFAs have the mutated amino acids as 1053 and 1078, should be 1054 and 1073

COMMENTS FROM THE ACADEMIC EDITOR:

Line 396 misegregation of nuclei and fig2a,

parental and kd dapi stain look the same in fig 2a. is dapi stain the measure of nuclear segregation?

line 418 centrosome shape suppl fig 3a

this is not convincing on the basis of one image and both parental and pp1 kd have elongated centrosomes

fig 3b labelling needs to be improved

---

## [Decision Letter · Decision Letter 2]

29 Jul 2024

Dear Dr Gissot,

Thank you for your patience while we considered your revised manuscript "The PP1 phosphatase exhibits pleiotropic roles controlling both daughter cell formation and amylopectin-steady state levels in Toxoplasma gondii." for publication as a Research Article at PLOS Biology. This revised version of your manuscript has been evaluated by the PLOS Biology editors and the original reviewers. Unfortunately, we were not able to contact the Academic Editor at this point, so it might be possible that we have some minor requests at the next stage. 

Based on the reviews, we are likely to accept this manuscript for publication, provided you satisfactorily address the remaining points raised by the reviewers, more specifically the textual comments from reviewer #2. Please also make sure to address the following data and other policy-related requests.

a) We routinely suggest changes to titles to ensure maximum accessibility for a broad, non-specialist readership, and to ensure they reflect the contents of the paper. In this case, we would suggest a minor edit to the title, as follows. Please ensure you change both the manuscript file and the online submission system, as they need to match for final acceptance:

"PP1 phosphatase controls both daughter cell formation and amylopectin levels in Toxoplasma gondii"

b) Thank you for providing the names of the foundations that supported the project. However, we also require the grant number (or grant titles if not available)

Please supply the numerical values either in the a supplementary file or as a permanent DOI’d deposition for the following figures:

Figure 1D, 2CD, 3C, 4CD, 6CD, S1BC, SDE, S3BD, S4AB

d) Please cite the location of the data clearly in all relevant main and supplementary Figure legends, e.g. “The data underlying this Figure can be found in S1 Data” or “The data underlying this Figure can be found in https://doi.org/10.5281/zenodo.XXXXX”

e) We require the original, uncropped and minimally adjusted images supporting all blot and gel results reported in the Figures:

Figure 1C, 5F, S1A

We will require these files before a manuscript can be accepted so please prepare and upload them now. Please carefully read our guidelines for how to prepare and upload this data: https://journals.plos.org/plosbiology/s/figures#loc-blot-and-gel-reporting-requirements

f) Please ensure that your Data Statement in the submission system accurately describes where your data can be found and is in final format, as it will be published as written there.

g) Per journal policy, if you have generated any custom code during the course of this investigation, please make it available without restrictions upon publication. Please ensure that the code is sufficiently well documented and reusable, and that your Data Statement in the Editorial Manager submission system accurately describes where your code can be found.

We expect to receive your revised manuscript within two weeks. 

*Published Peer Review History*

*Press*

Sincerely,

Melissa

Melissa Vazquez Hernandez, Ph.D.

Associate Editor

PLOS Biology

REVIEWERS' COMMENTS:

Reviewer #1: The authors addressed most of my comments. The manuscript is now clearer and improved.

Reviewer #2: 

The authors have addressed many of the comments from the previous version and it reads more focussed. The explanation of the differences to other studies published on PP1 in Toxoplasma is helpful. They have also improved the images adding expansion images and better labelling etc is provided. However I am bit confused with the previous study that they studied the role of PP1 in motility and egress whereas the phenotype here shows abnormal division, so did the author tried to study the PP1kd parasite for the e same phenotype as the previous study reported. Some discussion on it will be helpful. It is also not clear if the IMC and centrosome dynamics are linked 

 during cell division so why PP1 deletion affects only the IMC and not the centrosome division.

I have few minor comments :

1. The method section do not describe about the technique of expansion microscopy they have included in the revised version 

2. Line 101 may be add chromosome segregation rather than just segregation 

3. line 457 it should be "undergoing" instead of undoing mitosis 

Reviewer #3: 

The revised manuscript is much improved, satisfies the previous reviewer comments and is acceptable for publication.

---

## [Editor Report · Decision Letter 3]

7 Aug 2024

Dear Dr Gissot,

Thank you for the submission of your revised Research Article "PP1 phosphatase controls both daughter cell formation and amylopectin levels in Toxoplasma gondii." for publication in PLOS Biology. On behalf of my colleagues and the Academic Editor, Michael Duffy, I am pleased to say that we can in principle accept your manuscript for publication, provided you address any remaining formatting and reporting issues. These will be detailed in an email you should receive within 2-3 business days from our colleagues in the journal operations team; no action is required from you until then. Please note that we will not be able to formally accept your manuscript and schedule it for publication until you have completed any requested changes.

PRESS

Sincerely, 

Melissa

Melissa Vazquez Hernandez, Ph.D., Ph.D.

Associate Editor

PLOS Biology
